# Characterization of antibiotic resistance genes in the species of the rumen microbiota

Yasmin Neves Vieira Sabino[1], Mateus Ferreira Santana[1], Linda Boniface Oyama[2], Fernanda Godoy Santos[2], Ana Júlia Silva Moreira[1], Sharon Ann Huws[2]* & Hilário Cuquetto Mantovani [1]*

Infections caused by multidrug resistant bacteria represent a therapeutic challenge both in clinical settings and in livestock production, but the prevalence of antibiotic resistance genes among the species of bacteria that colonize the gastrointestinal tract of ruminants is not well characterized. Here, we investigate the resistome of 435 ruminal microbial genomes in silico and confirm representative phenotypes in vitro. We find a high abundance of genes encoding tetracycline resistance and evidence that the *tet*(W) gene is under positive selective pressure. Our findings reveal that *tet*(W) is located in a novel integrative and conjugative element in several ruminal bacterial genomes. Analyses of rumen microbial metatranscriptomes confirm the expression of the most abundant antibiotic resistance genes. Our data provide insight into antibiotic resistange gene profiles of the main species of ruminal bacteria and reveal the potential role of mobile genetic elements in shaping the resistome of the rumen microbiome, with implications for human and animal health.

[1] Departamento de Microbiologia, Universidade Federal de Viçosa, Viçosa, Minas Gerais, Brazil. [2] Institute for Global Food Security, School of Biological Sciences, Queen's University Belfast, Belfast, UK. *email: s.huws@qub.ac.uk; hcm6@ufv.br

Without immediate action to tackle the current and escalating threat of antimicrobial resistance (AMR), it is estimated that resistant infections may kill one person every 3 s by the year 2050, raising the death toll worldwide to 10 million annually[1]. This antibiotic-resistance crisis can be linked to several issues, including the overuse of these compounds for medical and agricultural purposes, inappropriate antibiotic prescribing, and limited discovery/development of novel and effective antibiotics[2]. The widespread use of antibiotics in the agricultural sector plays an underrepresented role in the AMR context. Large amounts of antimicrobials are frequently used in livestock to prevent diseases and promote animal growth[3]. More than 2 million kilograms of medically important antibiotics were sold in the USA for use in cattle in 2017, which corresponded to 42% of the total used in food-producing animals, and ruminants have been recognized as a potential reservoir of antibiotic-resistance genes (ARGs)[4,5].

The rumen ecosystem is colonized by a complex and genetically diverse microbiota, in which dense populations of microbes often exist in close proximity within biofilms[6]. Therefore, mechanisms that lead to horizontal acquisition of novel genes could provide competitive advantage for resource competition and facilitate the exchange of genetic material between members of the rumen microbiota and also with allochthonous species that occupy the same site in the ruminant gastrointestinal tract (GIT)[7–10]. In addition, feeding antimicrobials through ruminant diets can select for resistant organisms, potentially modifying the autochthonous ruminal microbiota[11]. These findings corroborate with recent analyses of the ovine rumen resistome, which identified resistances to 30 known antibiotics, including high abundance of genes for daptomycin and colistin resistance, two clinically relevant antibiotics[5].

Ruminants represent a major source of animal protein for human consumption worldwide (through milk and meat production), and cattle are raised in close proximity with humans, such as in rural areas, particularly in lower-middle-income nations, where a great proportion of the livestock farms are small, family-owned properties. Nonetheless, little is known about the occurrence and distribution of ARGs among the bacterial species of the core rumen microbiome. Recently, hundreds of reference genomes of cultured ruminal bacteria and archaea have been made available through the Hungate Project (http://www.hungate1000.org.nz/), representing ~75% of the genus-level bacterial and archaeal taxa present in the rumen[12]. We therefore hypothesized that genome mining of the Hungate genomic resources could reveal the distribution and genetic context of ARGs within major ruminal species represented in the core rumen microbiome.

As such we analyzed 435 genomes of ruminal bacteria and archaea to search for ARGs using different computational approaches. Identified ARGs were also evaluated for selective pressure, genetic organization, association with mobile genetic elements, as well as their expression in different rumen metatranscriptome data sets. The resistant phenotype was confirmed for some cultured representative ruminal bacteria from the sequenced Hungate1000 collection using in vitro testing and a novel integrative and conjugative element (ICE) associated with tetracycline resistance was identified in the genomes of rumen bacteria. Our findings shed light on the distribution and frequency of ARGs among the main species of bacteria colonizing the rumen, thereby improving our understanding of the mechanisms underlying antibiotic resistance within this microbial ecosystem.

## Results

**Detection of ARGs in ruminal microbial genomes.** The number and classes of ARGs detected in rumen microbial genomes using ResFinder[13], Resfams[14], and ARG-ANNOT[15] varied according to differences in specificity, sensitivity, and search method used by each computational tool. ResFinder identified a total of 141 acquired ARGs distributed in 72 rumen microbial genomes and across 11 antibiotic classes (Table 1). ARG-ANNOT detected 754 genes from 10 distinct antibiotic classes in 93 genomes of rumen bacteria, while Resfams predicted 3148 sequences related to the resistance of 9 classes of antibiotics in 430 rumen microbial genomes (Table 1; Supplementary Fig. 1). The most abundant ARGs detected by all three bioinformatic tools were related with resistance to beta-lactams (726 genes), glycopeptides (510), tetracycline (307), and aminoglycosides (193).

Resfams detected the majority of the aminoglycosides and glycopeptides-resistance genes, while ARG-ANNOT identified the most beta-lactam-resistance genes in the rumen bacterial genomes. Fluoroquinolone, fosfomycin, nitroimidazole, and trimethoprim were the antibiotic classes with the lowest number of ARGs identified using ResFinder, Resfams, and ARG-ANNOT. These putative ARGs were more prevalent among members of the phyla *Firmicutes*, *Proteobacteria*, *Bacteroidetes*, and *Actinobacteria*, but their abundances varied according to the bioinformatic tool used for ARG detection (Table 1; Supplementary Fig. 2).

**Phylogenetic distribution of ARGs.** Although ARGs were widely distributed across the genomes of ruminal bacteria (Fig. 1), resistance to specific antibiotic classes were more prevalent in some bacterial taxa, especially at the family and genus level.

### Table 1 In silico detection of ARGs in ruminal microbial genomes

| General features | ResFinder | ARG-ANNOT | Resfams |
|---|---|---|---|
| ARGs detected | 141 | 754 | 3148 |
| Genomes with ARGs | 72 | 93 | 430 |
| *ARGs by antibiotic class/function* | | | |
| Aminoglycosides | 10 | 26 | 157 |
| Beta-lactams | 11 | 446 | 269 |
| Colistin | 0 | 0 | 0 |
| Fosfomycin | 2 | 2 | 0 |
| Fluoroquinolones | 4 | 4 | 2 |
| Fusidic acid | 0 | 0 | 0 |
| Glycopeptides | 22 | 76 | 412 |
| MLS | 22 | 44 | 44 |
| Nitromidazole | 3 | 0 | 0 |
| Oxazolidinone | 0 | 0 | 0 |
| Phenicols | 3 | 10 | 19 |
| Rifampicin | 0 | 0 | 0 |
| Sulfonamides | 2 | 2 | 0 |
| Tetracyclines | 61 | 130 | 116 |
| Trimethoprim | 1 | 14 | 0 |
| ABC efllux | – | – | 1749 |
| RND efllux | – | – | 165 |
| Others[a] | – | – | 215 |
| *ARG's by taxonomic category* | | | |
| Actinobacteria (n = 32) | 7 | 11 | 164 |
| Bacteroidetes (n = 50) | 30 | 67 | 199 |
| Fibrobacteres (n = 2) | 0 | 0 | 4 |
| Firmicutes (n = 313) | 89 | 233 | 2502 |
| Fusobacteria (n = 1) | 0 | 0 | 2 |
| Proteobacteria (n = 22) | 15 | 443 | 244 |
| Spirochetes (n = 5) | 0 | 0 | 27 |
| Euryarchaeota (n = 10) | 0 | 0 | 6 |

*MLS* Macrolides, Lincosamides, Streptogramins
[a]ARGs that could not be ranked in any of the above antibiotic classes/functions; (—) these ARGs are not included in the ResFinder and ARG-ANNOT data sets

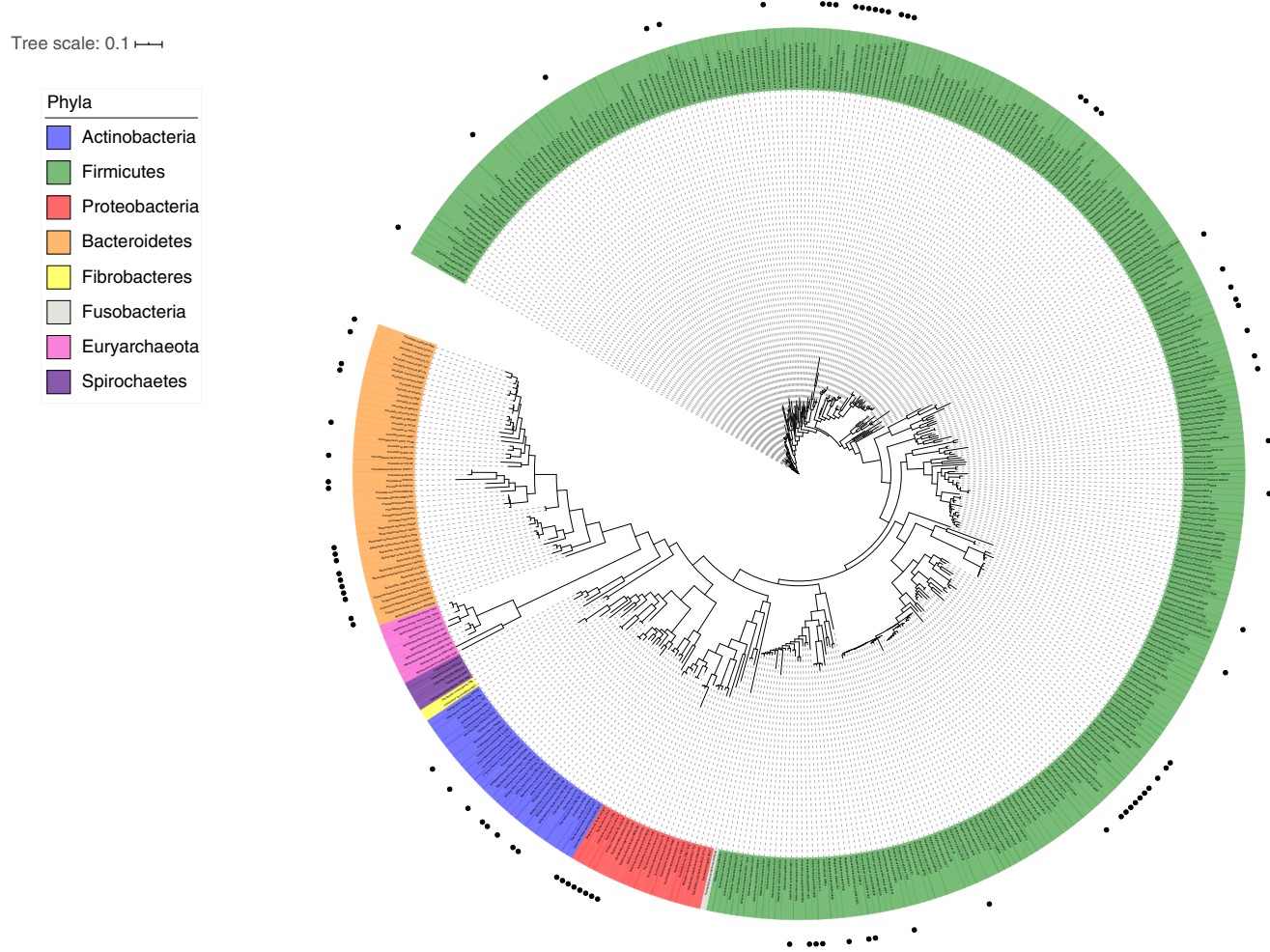

**Fig. 1** Antibiotic-resistance distribution in ruminal microbes. The 16S rRNA gene sequences used to construct the tree were obtained from the Hungate1000 collection and aligned using RDP Aligner. The phylogenetic tree was generated with the Maximum-Likelihood method using FastTree (1000 replicates), and visualized and annotated using iTOL. Black circles outside the tree represent the genomes, in which ARGs were identified from at least two bioinformatics tools applied in this study

Members of the *Proteobacteria* phylum showed a high proportion of genomes harboring resistance genes, with ARGs being detected in all genomes of the *Enterobacteriaceae* family. In the *Bacteroidetes* and *Actinobacteria* phyla, ARGs were concentrated in the genus *Bacteroides* and *Bifidobacterium*, respectively. In the *Firmicutes* phylum (which corresponded to the largest number of genomes analyzed in this study), resistance genes were detected across members of the *Lachnospiraceae* (*Lachnosclostridium*, *Clostridium*, and *Blautia*) and *Veillonellaceae* (*Selenomonas*, *Megamonas*, and *Megasphaera*) families, as well as in all species of the genus *Enterococcus* and among strains of *Staphylococcus epidermidis*. No antibiotic-resistance genes were detected in genomes belonging to the *Spirochetes*, *Fibrobacteres*, *Fusobacteria*, and *Euryarchaeota* phyla using Resfinder and ARG-ANNOT, but it should be noted that the number of genomes representing these taxa were much lower compared with the other phyla analyzed in this study. Although Resfams detected some potential ARGs in these microbial taxa, the majority were ABC efflux pumps. The ABC efflux pumps represent one of the largest protein families in microorganisms, contributing not only to reduce the intracellular concentrations of toxic compounds but also to the influx of substrates and other nutrients, and are not necessarily related to antibiotic resistance[16].

The distribution of the ARGs in ruminal microbial genomes was represented by the antibiotic class and according to the 16S

rRNA gene phylogeny (Fig. 2). Majority of the genomes (69.2%) showed resistance to one antibiotic class, with a dominance of tetracycline-resistance genes, which were distributed in bacterial genomes within the phyla *Actinobacteria*, *Firmicutes*, *Proteobacteria*, and *Bacteroidetes*. Phylogenetic analysis of the most abundant tetracycline-resistance genes (*tet*(W), *tet*(Q), and *tet*(O)) revealed that *tet*(W) was highly conserved across different taxa of ruminal bacteria, with a minimum sequence identity of 94.9 and 99% coverage in at least 28 ruminal bacterial genomes (Supplementary Figs. 3–5).

Moreover, as shown in Fig. 2, metronidazole and fosfomycin resistance were only identified in the genus *Prevotella* and in two strains of *Staphylococcus epidermidis*, respectively. *Staphylococcus epidermidis* together with *Citrobacter* sp. NLAE-zl-C269 were the only species harboring resistance genes to quinolones. *Escherichia coli* PA-3 was the single ruminal genome showing the highest number of ARGs (*n* = 5), including tetracycline resistance, beta-lactam resistance, aminoglycoside resistance, trimethoprim resistance, and sulfonamide resistance. *Bacteroides ovatus* NLAE-zl-C500 was the only species in *Bacteroidetes*-possessing genes for sulfonamide resistance. Beta-lactam-resistance genes were concentrated in the family *Enterobacteriaceae*, in the genus *Bacteroides*, and in two clades harboring the genera *Selenomonas*, *Staphylococcus*, *Succiniclasticum*, and *Bacillus*. *Proteus mirabilis* NLAE-zl-G534 and *Proteus mirabilis* NLAE-zl-G285 shared the

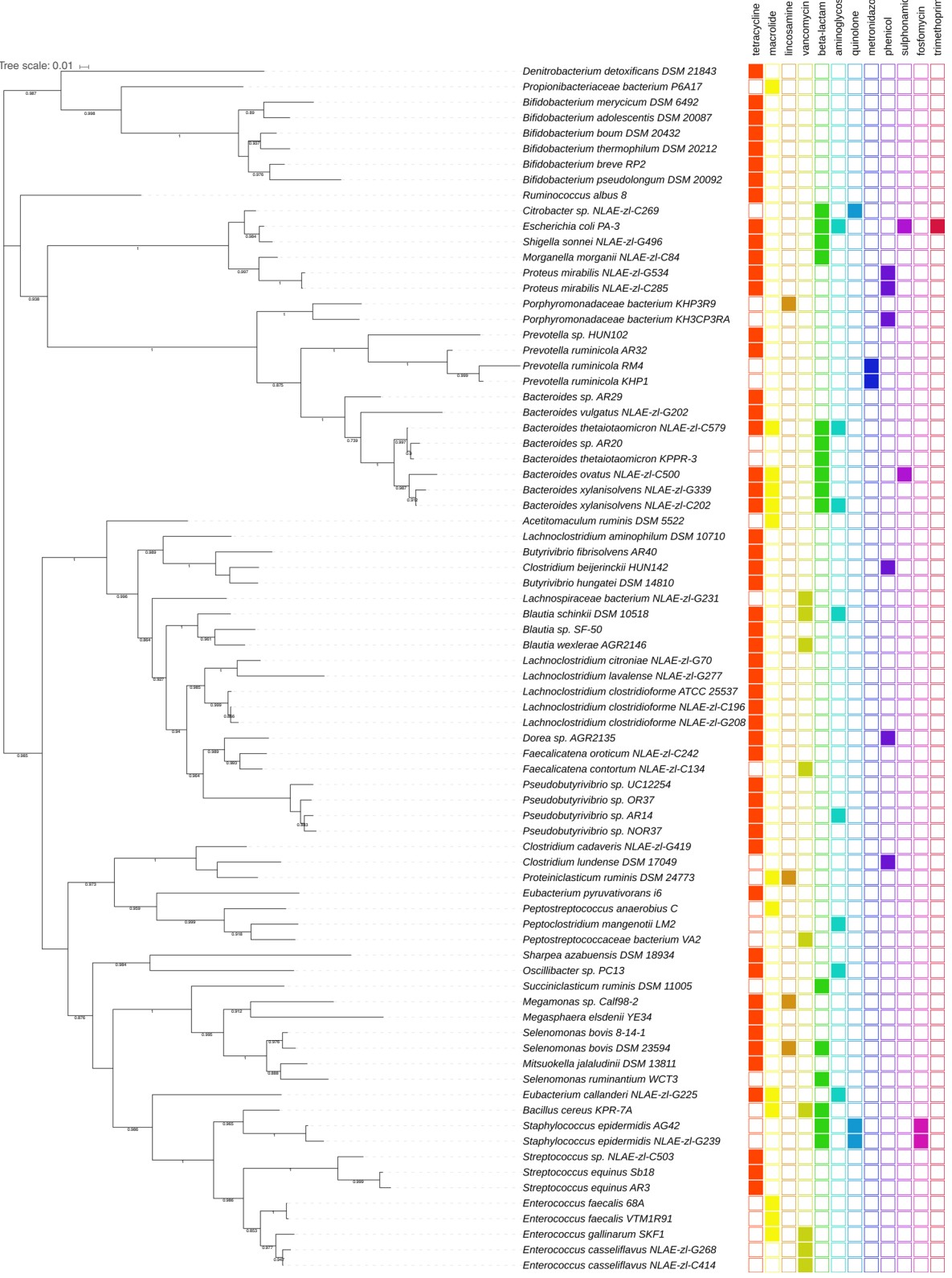

**Fig. 2** Distribution of ARGs in species of ruminal bacteria. The distribution of the resistance genes detected by at least two bioinformatic tools used in this study is presented in front of the tree as filled colored boxes. The 16S rRNA gene sequences used to construct the tree were obtained from the Hungate1000 collection and aligned using RDP Aligner. The tree was generated with the Maximum-Likelihood method using FastTree (1000 replicates) and visualized and annotated using iTOL. Only bootstrap values >0.7 are shown

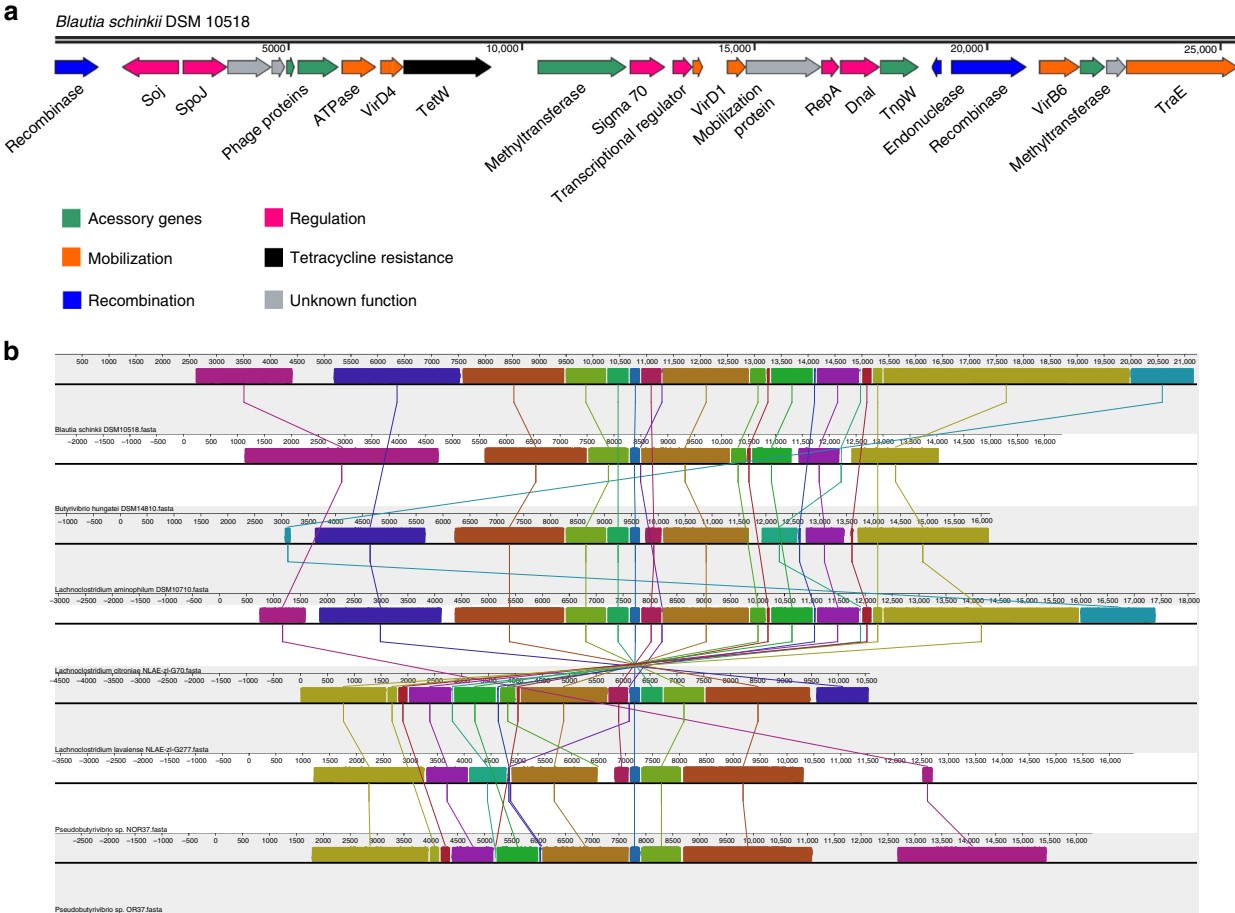

**Fig. 3** ICE_*RbtetW_07* structure and conservation. **a** ICE_*RbtetW_07* structure in *Blautia schinkii* DSM 10518. The genes of the ICE are represented by arrows classified and colored in modules according to their function. The essential modules of the ICE machinery are represented in orange, blue, and pink, for mobilization, recombination, and regulation genes, respectively. The ICE is located in scaffold 17 (nucleotide positions from 88079 to 111915) of the *Blautia schinkii* DSM 10518 genome. **b** Alignment of the ICE_*RbtetW_07* from ruminal bacterial genomes *Blautia schinkii* DSM 10518, *Butyrivibrio hungatei* DSM 14810, *Lachnoclostridium aminophilum* DSM 10710, *Lachnoclostridium citroniae* NLAE-zl-G70, *Lachnoclostridium lavalense* NLAE-zl-G277, *Pseudobutyrivibrio* sp. NOR37, and *Pseudobutyrivibrio* sp. OR37. Lines connecting blocks with identical colors represent the aligned regions, and show synteny or gene rearrangements. The *tet*(W) gene is represented in blue in the center of the figure

same pattern of resistance to tetracycline and phenicol, while *Staphylococus epidermidis* AG42 and *Staphylococus epidermidis* NLAE-zl-G239 harbored highly homologous resistance genes for beta-lactams, quinolones, and fosfomycin.

**ARGs genetic context**. The genetic locus of the *tet*(W) genes as well as their flanking regions were analyzed to investigate their mobility potential. For this, the 28 genomes harboring *tet*(W) genes detected by ResFinder were evaluated in sequences of up to 4000 bp within the scaffold where *tet*(W) was located. Genes encoding a putative methyltransferase protein were detected adjacent to or located immediately upstream or downstrem of the *tet*(W) gene in 18 genomes. From these, the gene encoding the protein under the accession number D1PK82 [https://www.uniprot.org/uniprot/D1PK82] was the most prevalent, and showed 98.9–100% sequence identity across the genomes of ruminal bacteria. In seven of these genomes, a gene encoding the Maff-2 protein, which is known to be involved in bacterial resistance to tetracycline, was also found flanking the *tet*(W) gene. Genes encoding transposon proteins (e.g., TnpW, TnpV, Tnp, TnpX, and ISPsy9) were identified flanking the *tet*(W) gene in 10 ruminal microbial genomes, and 22 out of the 28 genomes harboring *tet*(W) genes also had genes encoding proteins potentially

involved in bacterial conjugation, such as relaxases, plasmid mobilization proteins (e.g., MobC), and conjugation proteins (e.g., TraE, TraG/TraD). In addition, genes enconding phage proteins (e.g., PhiRv2) were detected in the flanking regions of the *tet*(W) genes in ten ruminal microbial genomes (Supplementary Data 1). Nonetheless, the scaffolds carrying the *tet*(W) genes lacked complete mobile elements showing the structural organization commonly found in plasmids, phages, and transposons.

A more detailed analysis of the sequences flanking the *tet*(W) gene revealed that at least seven genomes contained the complete machinery required for a functional ICE, despite the fact that no such genetic elements were found when the bacterial genomes were screened for using ICEberg[17]. The genetic organization of the ICEs identified in the genomes of ruminal bacteria, as exemplified for *Blautia schinkii* DSM 10518 (Fig. 3a), contained putative genes encoding for conjugation proteins (e.g., TraE), secretion proteins (e.g., VirD4), proteins controlling the cellular cycle (e.g., RepA, DnaI, σ70), and recombinases, in addition to other accessory proteins commonly found associated with these mobile elements (e.g., ATPases, methyltransferases, transposons, and phage-related proteins). The hypothesis that a novel ICE (named here as ICE_*RbtetW_07*) could be mediating tetracycline resistance in the genomes of rumen bacteria was reinforced by the presence of conserved elements (reversed organization in three

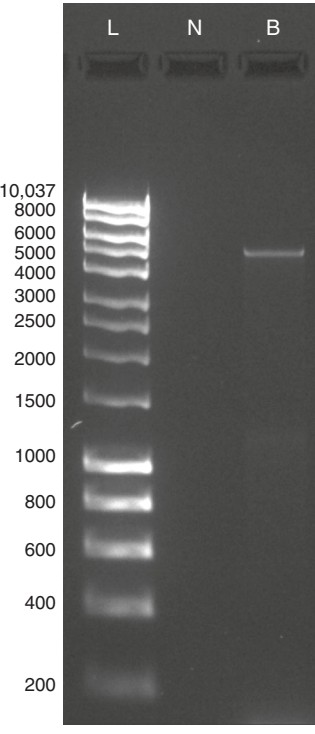

**Fig. 4** PCR detection of the ICE_RbtetW_07 in *Blautia schinkii* DSM 10518. The expected amplicon size of the region from the *virD4* gene to the *tet*(W) gene was 4794 pb. L: 1 kb HyperLadder, N: negative sample, and B: *Blautia schinkii* DSM 10518

genomes) in the ICE carrying the *tet*(W) genes on ruminal bacterial chromosomes (Fig. 3b). Moreover, phylogenetic analyses indicated that the ICE_RbtetW_07 had low sequence homology to other integrative conjugative elements previously reported (Supplementary Fig. 6).

**Detection of the ICE_RbtetW_07 in *B. schinkii* DSM 10518.** To confirm the presence of the ICE carrying the *tet*(W) genes in rumen bacteria, *Blautia schinkii* DSM 10518 was grown in the batch culture, and a fragment of 4794 bp covering the region from the *virD4* gene to the *tet*(W) gene was PCR amplified from the genomic DNA (Fig. 4). The amplicon was subjected to shotgun sequencing performed on the MiSeq with 300 cycles using the next-Flex Illumina workflow, and the assembled reads confirmed the genetic organization of the ICE (type IV secretion system gene adjacent to the tetracycline-resistance gene) predicted in the *Blautia schinkii* DSM 10518 genome (accession number PRJNA223460, [https://www.ncbi.nlm.nih.gov/bioproject/223460]) with 100% of nucleotide identity and 99.7% of coverage (Supplementary Data 2).

**Selection pressure analysis of *tet* resistance genes.** Because the tetracycline-resistance genes *tet*(W), *tet*(Q), and *tet*(O) were highly abundant and broadly distributed in the genomes of ruminal bacteria, we investigated if these genes are evolvable and if genetic variants are being selected for in the genomes of ruminal bacteria. For this purpose, the number of non-synonymous per synonymous substitutions ($d_N/d_S$ ratio) was calculated to determine the codon substitution rate, which could indicate the evolution of the gene in ruminal microbial genomes. Our analysis confirmed the presence of positive selection pressure in the *tet*(W) gene, with a $d_N/d_S$ ratio > 1 for 36 out of the 639 amino acid residues in the protein (Fig. 5a). Sites showing positive selection were mainly clustered in three distinct regions of the TetW protein, located between amino acids positions

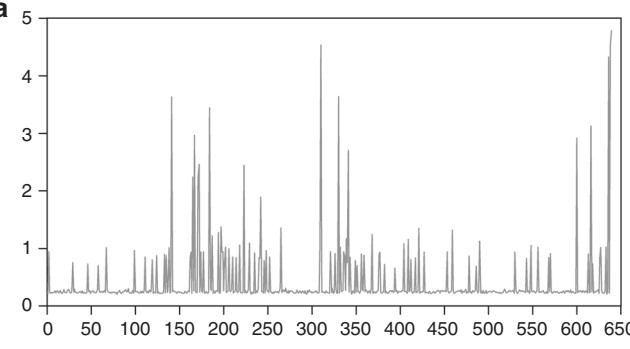

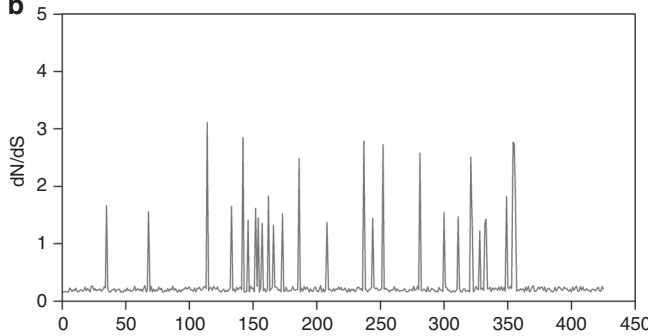

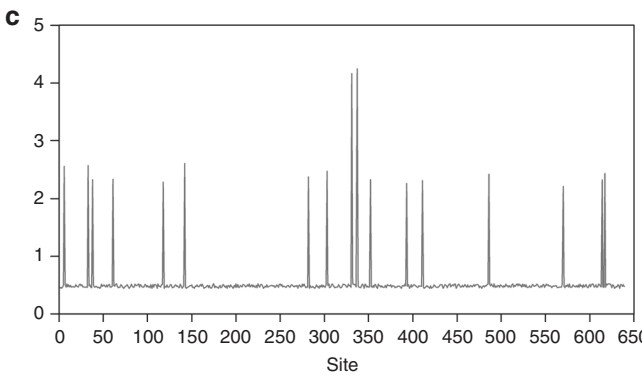

**Fig. 5** Selection pressure analysis in tetracycline-resistance genes. The $d_N/d_S$ ratio was calculated for the *tet*(W) (**a**), *tet*(Q) (**b**), and *tet*(O) genes (**c**) using JCoDA. The lines in each panel represent $d_N/d_S$ scores by site. The phylogenetic tree generated by JCoDA for $d_N/d_S$ analysis was reconstructed using the aligned protein sequences of the ARGs and Maximum-Likelihood approach with 100 replicates. Source data are provided as a Source Data file

141–265, 310–341, and 600–639, with an average $d_N/d_S$ ratio per site of 1.9, 2.6, and 3.1, respectively. For the *tet*(Q) and *tet*(O) genes, our analysis indicated 25 and 17 amino acid sites with positive selection pressure, respectively (Fig. 5b, c). Nonetheless, the overall $d_N/d_S$ ratio calculated for these genes was <1, and the likelihood ratio test data (LRT) for the substitution models did not present statistically significant difference at 95% confidence level, indicating absence of positive selection for *tet*(Q) and *tet*(O) genes. To confirm that the *tet*(W) gene is under positive selection, neutrality tests were applied in these ARG sequences. The results of Tajima's D ($D = -1.65$) and Fu and Li test ($D^* = -2.22$. $p < 0.05$; $F^* = -2.39$, $p < 0.05$) corroborate with our previous findings.

**Confirmation of resistance phenotypes.** To validate the resistance phenotype that was computationally predicted in the genomes of ruminal bacteria, 26 pure cultures matching the genomes

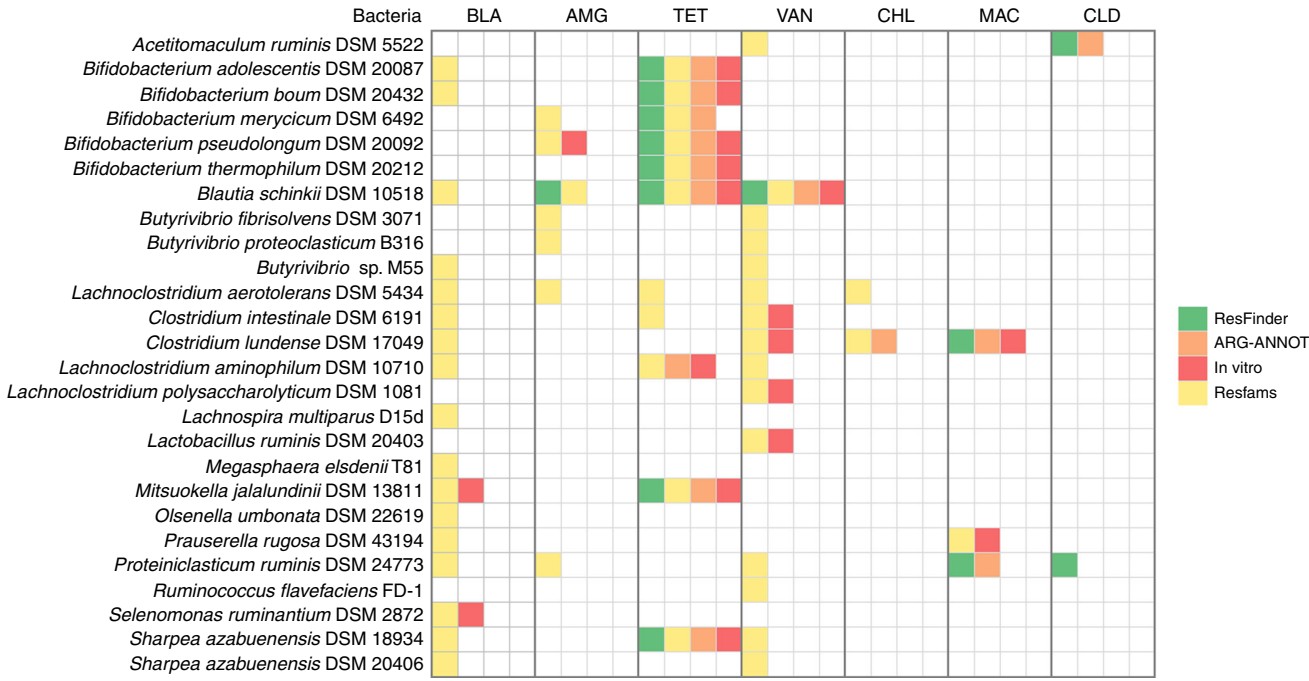

**Fig. 6** Concordance between antibiotic resistance detected in silico and in vitro. Twenty six bacteria were analyzed both in silico and in vitro in this study. Colors represent the approach used to predict the ARG or to detect the resistance phenotype of each bacteria. BLA (beta-lactam); AMG (aminoglycoside); TET (tetracycline); VAN (vancomycin); CHL (Chloramphenicol); MAC (macrolide); CLD (clindamycin)

analyzed in silico were selected for further in vitro characterization. Our in silico analysis predicted 57 resistance determinants in the ruminal bacteria that were selected for in vitro testing. Minimum inhibitory concentration (MIC values) were determined using the Epsilometer test (E-test) method[18], and 18 resistance phenotypes were confirmed in vitro (Supplementary Table 1). From this, eight ARGs (~44.0%) were detected/confirmed both in silico and in vitro by all approaches used in this study (Supplementary Fig. 7). ARG-ANNOT predicted 14 ARGs in the 26 cultures of ruminal bacteria tested in vitro, and 10 resistance phenotypes (71.4%) were confirmed in the E-test assays. ResFinder and Resfams predicted 14 and 53 ARGs in the bacterial cultures, and resistance to 9 (64.3%) and 17 (32.1%) of these antibiotics were confirmed phenotypically (Supplementary Fig. 7). Therefore, resistance phenotypes predicted by ResFinder and ARG-ANNOT agreed, for the most part, with the in vitro results, while Resfams predicted several resistance mechanisms that could not be confirmed by other computational or in vitro methodologies used in this study.

In addition, tetracycline resistance was detected simultaneously by at least three approaches in 69% of the genomes and bacterial cultures, providing further evidence that tetracycline resistance is widespread in the rumen ecosystem. Moreover, some resistance phenotypes were confirmed by our in vitro assays even when the presence of resistance determinants were predicted by only one bioinformatic tool. This was observed for vancomycin resistance in *Clostridium* and *Lachnoclostridium* strains and in *Lactobacillus ruminis* DSM 20403, for beta-lactams in *Mitsuokella jalalundinii* DSM 13811 and *Selenomonas ruminantium* DSM 2872, for aminoglycosides in *Bifidobacterium pseudolongum* DSM 20092 and for macrolides in *Prauserella rugosa* DSM 43194 (Fig. 6).

**Expression of ARGs in metatranscriptomes**. To confirm if the ARGs detected in ruminal microbial genomes are expressed in the rumen ecosystem, we aligned the genes conferring resistance to the five most abundant antibiotic classes (aminoglycoside, beta-lactam, macrolide, tetracycline, and vancomycin) detected in rumen bacterial genomes to 15 rumen metranscriptome data sets of dairy and beef cattle and sheep. At least one resistance gene from each antibiotic class was expressed in one or more metatranscriptomes (Fig. 7). In general, tetracycline-resistance genes presented the highest level of expression, mainly the *tet*(O), *tet*(Q), *tet*(W), and *tet*(37) gene. Moreover, individual genes varied in their expression levels between different data sets, being more prevalent in sheep and beef cattle. Some of the ARGs identified in the ruminal microbial genomes, such as *strA*, *cepA*, *ermB*, *tet*(B), and *vanC*, among others, were not expressed in any of the metatranscriptome data sets investigated in this study (Fig. 7).

## Discussion

The hypothesis that livestock animals represent a reservoir of ARGs is not new, and previous studies have demonstrated that bacteria isolated from fecal samples of monogastrics and ruminants harbor genetic elements that can confer resistance to clinically used antibiotics[19]. However, less attention has been given to the occurrence of ARGs in the rumen ecosystem and their potential to be transferred to commensal and pathogenic bacteria. Microorganisms colonizing the rumen also disseminate to the environment through animal saliva during rumination and due to the flow of rumen microbial biomass to the omasum, abomasum and to the distal parts of the GIT, being released through fecal discharge into the soil[20].

In this work, we investigated the occurrence and distribution of ARGs in hundreds of ruminal microbial genomes made available through the Hungate1000 project[12]. Our results indicate that ARGs are widely distributed among ruminal bacteria, with genes conferring resistance to tetracycline being frequently detected in the genomes of several species. The regions flanking the tetracycline-resistance genes showed a conserved pattern in an apparently novel integrative and conjugative element (ICE), suggesting dissemination of antibiotic resistance through horizontal gene transfer in the rumen microbiome. In addition, we

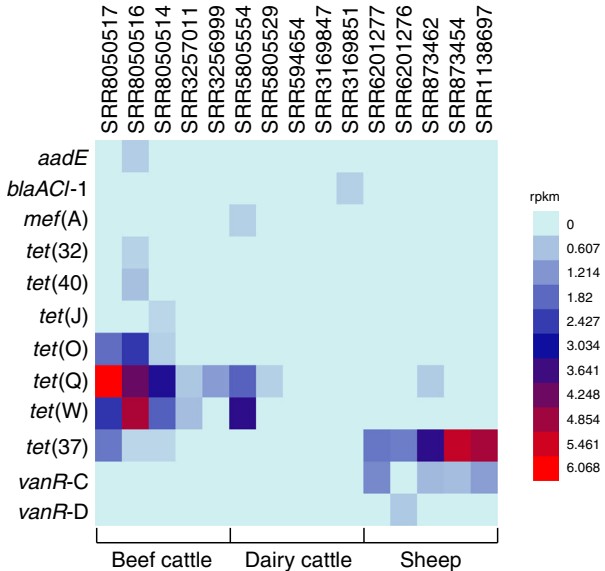

**Fig. 7** Expression levels (RPKM) of ARGs in ruminal metatranscriptomes. Beef and dairy cattle and sheep data sets were analyzed, and only ARGs that were expressed in at least one metatranscriptome are represented. Source data are provided as a Source Data file

show evidence of positive selective pressure on the *tet*(W) gene, which may contribute to the selection of genetic variants of this gene, conferring ribosome protection against tetracycline antibiotics in ruminal bacteria.

Tetracyclines are broad-spectrum antibiotics that inhibit protein synthesis in susceptible bacteria[21] and have been used in human and veterinary medicine to fight bacterial infections and to promote the growth of food-producing animals, improving feed efficiency and animal performance[22]. The growth-promoting activity of tetracyclines in livestock was first reported in 1949 when feeds supplemented with dried *Streptomyces aureofaciens* mash were administrated to poultry[23]. However, the long-term use of subtherapeutic doses of tetracyclines has been associated with increased levels of bacterial resistance in the GIT of food-producing animals[24], which led countries such as the USA and members of the European Union to ban the use of tetracyclines as growth promoters since 1975[25]. The prevalence of tetracycline-resistance genes among ruminal bacteria has a potential negative impact in livestock health given that tetracyclines account for up to 24% of the antibiotics used to treat respiratory, uterine, or locomotion diseases in Europe[26].

In our study, *tet*(W), *tet*(Q), and *tet*(O) were the main tetracycline-resistance genes identified by ResFinder, but *tet*(W) was the most abundant gene, being distributed in 28 genomes of ruminal bacteria with high similarity of nucleotide sequences. Since ResFinder is an online platform that uses whole-genome sequencing data to identify acquired AMR genes in bacteria[13], the *tet*(W) gene was selected for further analysis with an aim to investigate if it could be horizontally transferred among ruminal bacteria.

Our analyses suggest that the *tet*(W) gene of several species of ruminal bacteria are located on an ICE with a unique modular pattern, not previously reported for tetracycline resistance (ICE_*RbtetW_07*). The organization of the ICE structure is characterized by a core region containing essential modules for conjugation, recombination, and regulation[27]. The conjugation machinery often contains relaxases and secretion systems to mediate the transfer of DNA between donor and recipient cells. The recombination module usually contains genes encoding integrases,

excisionases, and recombinases, while genes encoding proteins involved in DNA transcription are found in the regulation module. Moreover, several accessory genes, including methyltransferases, transposons, phages, and plasmid related proteins, in addition to the ARGs, can be present in an ICE structure[27]. At least one essential gene from each module and several accessory genes were detected in the flanking region of the *tet*(W) gene in seven ruminal microbial genomes analyzed in this study, suggesting that ICEs could be mediating the horizontal transfer of tetracycline-resistance genes between ruminal bacteria. Some essential genes of integrative and conjugative elements were also found in other genomes of rumen bacteria harboring the *tet*(W) gene, but the genetic structure of these elements appeared incomplete. The structure of ICE_*Rb-tetW_07* was conserved in the genomes of ruminal bacteria, and the presence of the *tet*(W) gene on the ICE was confirmed in the genome of *Blautia schinkii* DSM 10518 by PCR amplification and shotgun sequencing. ICEs are known to be frequently transferred between distant bacterial taxa[28], and our analysis suggest that these genetic elements might have great mobility in the rumen ecosystem. If this is indeed the case, there is a potential threat of these genetic elements to be transferred to bacterial pathogens in the GIT of ruminants and into the environment.

Some groups of microorganisms, such as the *Enterobacteriaceae*, are frequently associated with the horizontal transfer of ARGs[29,30]. The enrichment of ARGs appears to be a common feature in the mobile resistome of Proteobacteria from both the animal and the human gut microbiome[31]. This may be associated with the frequent use of antibiotics targeting *Enterobacteriaceae* to control infections in the respiratory, urinary, and GIT of humans and livestock, thus selecting for resistant strains[5]. Our in silico results demonstrated that members of the Proteobacteria phylum that inhabit the rumen can harbor multiple ARGs, as exemplified by the genome of *E. coli* PA-3, a ruminal isolate that contains five genes conferring resistance to four distinct antibiotics. Although generic *Escherichia coli* populations are often not so abundant in the rumen (<10^6 cells/ml), cattle are considered a natural reservoir for pathogenic strains of *E. coli*[32,33]. In addition, management and nutritional practices, such as feeding the animals a high-grain diet, can cause significant increases in ruminal and fecal populations of *E. coli* and influence the abundance and diversity of ARGs in the ruminal resistome[33,34].

Furthermore, a complete operon for vancomycin resistance (*vanC*) was found in the genomes of *Enterococcus casseliflavus* and in *Enterococcus gallinarum* SKF1, which makes these species intrinsically resistant to this antibiotic[35]. Vancomycin is one of the last-option antibiotic used to treat infections caused by Gram-positive bacteria[36], but little is known about vancomycin resistance in ruminal bacteria. Our analyses predicted resistance to vancomycin in *Clostridium*, *Lachnoclostridium*, and *Blautia*, and in vitro experiments confirmed these phenotypes in *Blautia schinkii* DSM 10518, *Lachnoclostridium aerotolerans* DSM 5434, *Clostridium intestinale* DSM 6191, and *Clostridium lundense* DSM 17049. However, our transcriptomic analysis indicated that the expression of vancomycin-resistance genes in the rumen microbiome is low. In addition, vancomycin has limited use for therapeutic purposes in animals, and vancomycin analogs have been restricted as growth promoters in livestock[36].

Overall, our in vitro assays confirmed 31.6% of the resistance phenotypes predicted by computational approaches. Nonetheless, 64.3% and 71.4% of the resistance phenotypes based on the ARGs predicted by ResFinder and ARG-ANNOT, respectively, were confirmed in vitro. These results show concordance between these different approaches, despite the fact that some predicted ARGs might not be functional in vitro. In addition, our transcriptomic analyses showed higher expression of tetracycline-

resistance genes and low expression of other classes or ARGs (such as beta-lactams and aminoglycosides) in the rumen data sets, which also agreed with the resistance phenotype observed in vitro for the pure cultures of ruminal bacteria. Therefore, the results presented here, based on genomic, transcriptomic and phenotypic data, increase the confidence that ARGs are distributed among different species of ruminal bacteria, and that these genes are expressed by members of the microbial community in vivo. Nonetheless, additional studies will be needed to evaluate the effects of animal ageing, production systems, and nutritional practices, including antibiotic feeding, on the expression, and potential transit of the ARGs detected in this study across the GI tract of cattle.

Taken together, our results provide insights that characterize the antibiotic resistance in species of ruminal bacteria. Our findings demonstrate that ARGs, in particular the genes for tetracycline resistance, are prevalent among ruminal bacteria and subjected to positive selection pressure. Moreover, analysis of the flanking regions of the tet(W) genes provided evidence that this gene can be disseminated horizontally through ICE transfer. Given the fact that the rumen ecosystem also harbor a highly dense and diverse microbiota that maintain complex ecological associations, and the capacity of microorganisms to exchange DNA through direct (conjugation) and indirect mechanisms (transformation, transduction), the GIT of ruminants should be considered as a relevant source of antibiotic-resistance genes. The evidence of a new ICE (ICE_RbtetW_07) carrying the tet(W) gene emphasizes this idea and presents a risk for both animal and human health, due to the capacity of these genetic elements to disseminate into the environment. This hypothesis is also supported by the observation of non-phylogenetically related bacteria harboring the same resistance genes, and the expression of the same ARGs in ruminal metatranscriptomes data sets from beef and dairy cattle and sheep. In addition, our results highlight the importance of using multiple computational tools to search for ARGs in genomic data and demonstrate the relevance to validate these results in vitro and/or in vivo.

## Methods

**Collection of genomic data and identification of ARGs.** The sequence files of 435 ruminal microbial genomes (425 bacteria and 10 archaea), from the Hungate1000 project, were downloaded in FASTA format from the NCBI (National Center for Biotechnology Information, http://www.ncbi.nlm.nih.gov/genome) and the JGI (Joint Genome Institute, http://genome.jgi.doe.gov) websites (Supplementary Data 3). Putative ARGs in the bacterial and archaeal genomes were predicted using three bioinformatics tools (ResFinder v2.1, Resfams v1.2, and ARG-ANNOT Nt V3, see below) that vary in sensitivity and specificity and apply distinct databases and search methods to detect putative ARGs in nucleotide sequences.

We initially prospected the bacterial and archaeal genomes using the ResFinder v2.1 database (https://cge.cbs.dtu.dk/services/ResFinder/)[13] against all antibiotic classes available for analysis of acquired ARGs, which included aminoglycosides, beta-lactams, colistin, fluoroquinolones, fosfomycin, fusidic acid, glycopeptides, MLS (macrolides, lincosamides, and streptogramins), nitroimidazole, oxazolidinone, phenicols, rifampicin, sulfonamides, tetracyclines, and trimethoprim. For homology-based screening, the minimum gene identity and sequence length were set to 70 and 60%, respectively, in relation to the reference-resistance gene.

The software Resfams v1.2 was used to search for conserved structural domains of antibiotic-resistance function in protein sequences of ruminal bacteria[14]. The protein sequences of the analyzed genomes were downloaded in the FASTA format from the NCBI or JGI websites and when these sequences were not found, the online interface Prodigal v1.2 (http://compbio.ornl.gov/prodigal/server.html) was used to convert nucleotide to protein sequences using the Standard Bacteria/Archaea translation table code.

Genomes of ruminal microorganisms were also analyzed using the ARG-ANNOT database (http://en.mediterranee-infection.com/article.php?laref=283&titre=arg-annot-)[15]. Sequences of the ARGs in the ARG-ANNOT database (ARG-ANNOT Nt V3) were used for similarity analysis against the ruminal microbial genomes using BLASTn[37] on the Galaxy platform[38]. The alignment parameters were minimum sequence identity of 70%, sequence length cutoff of 60%, and E-value < 10$^{-6}$.

**Phylogenetic analysis and distribution of ARGs.** To evaluate if the ARGs grouped according to the evolutionary history of the microbial species, phylogenetic trees were reconstructed using the 16S rRNA gene sequences from the genomes analyzed in this study. The 16S rRNA gene sequences were obtained from the Hungate1000 collection[12]. Next, all the sequences were organized in a single txt file and aligned using the RDP Release 11.5 aligner in the Ribosomal Database Project website (RDP, https://rdp.cme.msu.edu/)[39]. FastTree v2.1 (http://www.microbesonline.org/fasttree/)[40] was used to infer Approximately-Maximum-Likelihood phylogenetic trees using default settings. The generated output file (.tree) was visualized and annotated on the Interactive Tree of Life (iTOL) interface v4 (https://iTOL.embl.de/)[41].

To evaluate sequence conservation of the ARGs that were identified in silico as described above, the sequences of the most frequent genes conferring resistance were extracted from the ruminal microbial genomes, and aligned as described below. To extract the sequences of the resistance genes, all genomes were annotated by Prokka v1.12[42] using the Galaxy platform (minimum contig size of 200 and E-value < 10$^{-6}$). To obtain the location of the resistance genes in the bacterial genomes, the ResFinder v2.1 database was aligned against the annotated genomes using the BLASTn tool on the Galaxy platform. Minimum query coverage and sequence identity were 60 and 70%, respectively, with a threshold E-value < 10$^{-6}$. The sequence of the resistance genes identified in the previous alignment were extracted using the SAMtools software v1.9 (http://SAMtools.sourceforge.net/)[43], and the most abundant ARGs were aligned using MUSCLE version 3.8.31[44]. The phylogenetic reconstruction was based on the Maximum-Likelihood method using FastTree v2.1, and iTOL v4 graphical interface was used to represent the phylogenetic trees, as described above.

**Genetic context of ARGs.** To assess potential mechanisms of ARG mobility, the genetic elements flanking the resistance genes in a scaffold were investigated using the software Artemis[45]. When these flanking elements were annotated as hypothetical protein, sequence function was predicted on the UNIPROT website (http://www.uniprot.org/)[46], using BLASTn, UniProtKB as the target database and default settings. Putative mobile genetic elements flanking the resistance genes were searched using different computational tools. ISfinder (https://www-is.biotoul.fr/)[47] was used to screen for transposons and/or integrons, and phage sequences were confirmed using PHASTER (http://phaster.ca/)[48]. The genomes that harbored proteins with predicted plasmid functions near the resistance genes were further analyzed against the Enterobacteriaceae and Gram-positive database of PlasmidFinder 2.0 (https://cge.cbs.dtu.dk/services/PlasmidFinder/)[49], using a threshold for minimum sequence identity of 70% and minimum coverage of 60%. Finally, ICEberg 2.0 (http://db-mml.sjtu.edu.cn/ICEberg/)[17] was used to identify ARGs carried by integrative and conjugative elements (ICE). Scaffolds harboring selected ARGs were used as query sequences (FASTA format) in the WU-BLAST2 search tool in ICEberg, using BLASTn as the search program and ICE nucleotide sequence as the sequence database. All the other parameters were kept as default settings.

**Conservation and in vitro confirmation of a novel ICE.** To evaluate the genetic conservation of the integrative and conjugative elements identified in the genomes of ruminal bacteria, the nucleotide sequences of these ICEs were aligned using progressiveMAUVE[50]. In addition, a phylogenetic tree was reconstructed in MEGA X software using Muscle to compare the sequences of the novel ICE with other ICEs available in the ICEberg database. Phylogenetic reconstruction was performed using the Maximum-Likelihood method with 100 replicates.

To confirm the presence of a novel ICE in the genome of ruminal bacteria, PCR amplifications were performed using Blautia schinkii DSM 10518. Primers were designed to amplify a 4794 pb DNA fragment that included the region from gene virD4 to gene tet(W). Briefly, the bacteria was grown in anaerobic media at 39 °C for 72 h. Genomic DNA was extracted using the Qiagen DNA extraction kit (Manchester, UK). PCR amplification of the region of interest was carried out in 25 μl volumes as follows: 12.5 μl 2x MyTaq HS Red Mix, 1 μl each of 20 μM primers drawn in this study (RbtetW07_F: ATGAAAAAGCAGCTTGACATCAAAAAGC and RbtetW07_R: TTACATTACCTTCTGAAACATATGGCGC), 8 μl of nuclease-free water, and 2.5 μl of 50 ng template DNA. The cycling conditions were as follows: initial denaturation step at 95 °C for 60 s, 35 cycles of denaturation at 95 °C for 15 s, annealing at 58 °C for 15 s, and extension at 72 °C for 3 min, and a final extension step at 72 °C for 5 min. The size and integrity of PCR products were visualized and assessed by agarose gel electrophoresis using a 1% (w/v) agarose in TAE buffer (40 mM Tris, pH 8.0, 20 mM acetic acid, 1 mM EDTA, BioRad Ltd., Hemel Hempstead, UK). The PCR product was subjected to shotgun sequencing performed on the MiSeq with 300 cycles using the next-Flex Illumina workflow at the Queen's University of Belfast Genomics Core Technology Unit (Belfast, UK) to confirm the presence of the predicted integrative and conjugative element. Quality filtering of the reads was performed using Trimmomatic software v0.27[51], and sequences were assembled using SPAdes 3.10.1[52]. The sequence was aligned to the ICE coding sequence in the genome of Blautia schinkii DSM 10518, using Clustal Omega[53].

**Selective pressure on identified ARGs.** Sequences of the most prevalent ARGs detected in ruminal microbial genomes were selected to evaluate if these genes are

under selection pressure. Selective pressure on the ARGs was estimated by the number of non-synonymous and synonymous substitutions ($d_N/d_S$) using the software JCoDA[54]. JCoDA uses ClustalW to align the nucleotide sequences and Phylogenetic Analysis using Maximum Likelihood (PALM) to calculate the $d_N/d_S$ ratio. Analyses were performed using nucleotide sequences in the FASTA format as input sequences and the universal (default) genetic translation code. The alignment option was set to ClustalW, and phylogenetic trees were reconstructed by the Maximum-Likelihood method with 100 replicates using protein sequences in the Phylip-sequential format. The $d_N/d_S$ was calculated by site (amino acid) using the Bayes Empirical Bayes (BEB) model, which takes into account the uncertainties of model parameters, avoiding the generation of false positives[54]. The selective pressure on the antibiotic-resistance genes were confirmed by two neutrality tests, the Tajima test (performed in MEGAX[55]) and Fu and Li test (DnaSP v6[56]).

**In vitro analysis of the predicted resistance phenotypes**. To evaluate if the ARGs detected in the ruminal microbial genomes conferred the predicted resistance phenotype in vivo, 26 cultures of ruminal bacteria that had their genomes analyzed for the presence of ARGs were tested in vitro to confirm these phenotypes. Cultures stored at −80 °C were activated twice in brain heart infusion (BHI) broth at 39 °C for 24 h under anaerobic conditions before each experiment. For the antibiotic susceptibility test, the optical density ($OD_{600nm}$) of the cultures was adjusted to obtain a cell count of ~$10^8$ CFU ml$^{-1}$ (equivalent to 0.5 McFarland scale). Subsequently, cultures were spread on the surface of solid BHI media using swabs. E-test strips (Biomérieux, Basingstoke, UK; Fannin, Dublin, Ireland) of the antibiotics to be tested were distributed on the surface of the inoculated medium, and the plates were incubated anaerobically at 39 °C for another 24 h. Only antibiotics to which resistance was predicted in silico were evaluated in vitro using pure cultures of ruminal bacteria. The minimum inhibitory concentration (MIC) of each antibiotic was determined at the intersection of the inhibition zone with the E-test strips[57]. All experiments were performed with three biological replicates. Reference values from the European Committee on Antimicrobial Susceptibility Testing (EUCAST) breakpoint table were used to interpret the MIC results[58]. In the case of a few antibiotics where breakpoint values were not available for the tested species, MIC interpretations were based on breakpoint values published for the most closely related organism.

**Metatranscriptomic analyses**. The metatranscriptome data sets were downloaded in the FASTQ format from the NCBI Sequence Read Archive SRA (https://www.ncbi.nlm.nih.gov/sra)[59]. Fifteen ruminal data sets representing cattle from distinct production systems (dairy and beef) and from sheep were selected for the analysis (Supplementary Table 2). FastQC software v0.11.5 (http://www.bioinformatics.babraham.ac.uk/projects/fastqc) was initially applied for quality control checks on raw sequence data using a Phred quality score of 20 to filter high-quality bases. However, because most reads were discarded using this criteria, gene expression was evaluated using raw sequence data.

In this study, the most abundant group of ARGs identified in the ruminal microbial genomes was selected to verify their expression in the metatranscriptomic data sets. These genes were predicted to encode resistance to aminoglycosides (*aadE*, *ant*(6)-Ia, *aph*(3')-III, *strA*, *strB*), beta-lactams (*cepA*, *blaACl*−1, *blaSED1*, *blaDHA*-2, *blaZ*), macrolides (*ermB*, *erm*(G), *mef*(A), *lsa*(A), *lsaC*), tetracycline (*tet*(32), *tet*(40), *tet*(A), *tetA*(P), *tet*(B), *tetB*(P), *tet*(D), *tet*(J), *tet*(M), *tet*(O), *tet*(Q), *tet*(W), *tet*(37), and vancomycin (*vanC*, *vanD*, *vanH*-D, *vanR*-C, *vanR*-D, *vanS*-C, *vanS*-D, *vanT*-C, *vanX*-D, *vanXY*-C, *vanY*-D, *vanZ*-F). The Bowtie2-build tool was used to index the DNA sequences of the resistance genes, while Bowtie2[60] was employed to align these sequences to the metatranscriptomic data sets. The expression level of each gene was calculated by the number of uniquely mapped reads per kilobase in a gene per million mappable reads (RPKM)[61] using a cutoff value of 0.3[62].

**Reporting summary**. Further information on research design is available in the Nature Research Reporting Summary linked to this article.

## Data availability

The data that support the findings of this study are included in the paper and/or its Supplementary Information files. Data underlying Figs. 5, 7 and Supplementary Figs. 1, 2, and 7 are provided as a Source Data file.

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

## Acknowledgements

This work has been supported by the Coordenação de Aperfeiçoamento de Pessoal de Nível Superior (CAPES; Brasília, Brazil), Fundação de Amparo a Pesquisa do Estado de Minas Gerais (FAPEMIG; Belo Horizonte, Brazil), and the Conselho Nacional de Desenvolvimento Científico e Tecnológico (CNPq; Brasília, Brazil). We are also grateful for RCUK Newton Institutional Link Funding (172629373).

## Author contributions

Y.N.V.S., H.C.M., and S.A.H. conceived the project. Y.N.V.S and A.J.S.M. completed the laboratory work under the supervision of H.C.M. and S.A.H. Y.N.V.S., M.F.S., and F.G.S. analyzed the data and generated figures. S.A.H. provided the anaerobic cultures analyzed in the study. L.B.O. assisted Y.N.V.S. with the in vitro experiments. L.B.O., M.F.S., and S.A.H. provided valuable ideas into the project from the time of conception. Y.N.V.S. and H.C.M. wrote the paper with imput from all co-authors.

## Competing interests

The authors declare no competing interests.
