## [Peer Review file · Nature Communications]

Reviewers' comments:

Reviewer #1 (Remarks to the Author):

Reviewer's comments

The manuscript titled "Selection pressure and mobile genetic elements shape the antibiotic resistome in the rumen microbiome" applied bioinformatic based approaches to identify ARGs from the known microbial genomes and have applied some validation approaches to confirm them. Although the concept of identifying ARGs using different bioinformatic tools is not new and novel, the authors made good efforts with the aim to verify the presence of ARGs in the rumen microbial cultures and/or rumen microbiome. But there are some major weaknesses on the current version, which prevents its publication at NC.

1. Many conclusions drawn without proper data support. One of the issues is the lack of proper experimental design to generate those data:

i). The authors applied three in silico tools to identified ARGs from 435 genomes of rumen bacteria and archaea, and reported significant different results among these methods (Pages 4-6). It is very common that different bioinformatic tools usually generate varied outcomes for the same sequence dataset. Here, it is extremely important that authors should consider the "common findings" identified from all three methods to be likely the "true" ARGs. Based on Figure 7, the AR in vitro assay also showed similar trend with the ARG identified from all three methods.

ii). The true and biological relevant functions of identified ARGs are not verified. In addition, the selection pressure and mobile genetic elements were only done based on "dry" analysis and lack of biological relevance. Need validation assay or another approach to verify the results:

For example, comparison the rumen resistomes between cattle fed with antimicrobials vs without antimicrobials or fed with antimicrobials and then withdraw the antimicrobials to validate the selection pressure (either positive or negative) on rumen resistomes. Similarly, the mobile genetic elements should be verified in the true data set. In addition, the authors calculated dn/ds based on genomes. Can such value be validated in the in vitro assays?

iii). The comparison between rumen microbiome and human fecal microbiome does not make any sense. The overall introduction, discussion and conclusions drawn on the linkage between rumen resistome and human AMRs is vague and lack of data support.

Although it is trendy to make the linkages between livestock resistome with human AMR concerns, the conclusion can not be just drawn based on one beef cattle rumen metatranscriptome, one dairy cattle rumen metatranscriptome and one human fecal metatranscriptome. If authors aim to identify the "evidence" between rumen and human, samples should be collected from cattle rumen, fecal,

soil and the fecal from the person who have contact with cattle. With only three random datasets, it does not provide data that support the conclusion.

2. Need more stringent cut-offs and more datasets. It is not clear how the cut offs (70% identify and 60% coverage) were determined (pages 24 and 25). How do authors know these are strong enough? What is the cut-off for RKPM for the gene expression? Based on Table 1, the RKPM values are low (0.01-0.1) (6 out of 8 ARGs) for beef cattle rumen metatranscriptome which should be considered as undetectable. For gene expression, the RKPM should be higher than 0.2 or 0.3 to be considered as expressed. Similar, (6 out of 10) in dairy rumen metatranscriptome should be also not expressed.

3. The focus on tetracycline resistance is more computer bases, not biology driven.

i). tet(w) is identified in a couple of Bifidobacterial species, what does this mean?

ii). Also, the authors need to clearly indicate how and when the tetracycline has been admitted in the ruminant production and how the identified tet(s) are relevant to the ruminant production system.

iii). Again, the linkage to human is very weak.

Other comments:

Through the manuscript, it should be ruminal microbial genomes, not ruminal genomes. One refers to microbes, the other refers to host genome.

Suggest moving the validation using metatranscriptomes section to be the end of results to make the flow better.

Page 2: the first paragraph is not very relevant, suggest to simplify

Second paragraph, please make all statements to be more specific especially about the AM use in livestock animals. Here should include information on ruminants and limit the cite the information from swine and poultry production.

Pages 4-6: very confusing. It would be better if authors can summarize the outcomes from three approaches using a Table, since the outcome of the comparison among three methods is one of the conclusions.

Page 7 and page 22: the ARG from E. coli PA-3: It has been well known varied E. coli strains carry ARGs. The authors should here indicate how this strain differ from other strains? Also, does this train specific to rumen? Could this be an environmental organism?

Page 9: not clear what is the purpose to draw the network and what do these networks suggest?

Suggest to significantly reduce the discussions on human microbiome, really focused on rumen microbiome and maybe ruminant cattle microbiome.

Page 19: statement (tet(w) gene in butyrivibrio fibrisolvens) based on citation no. 30. No such resistance from this manuscript (figure 7). What does this suggest?

Page 20: more data are needed to support the conclusions drawn in the first paragraph.

The first part of 2nd paragraph is also vague and not relevant to the current study, and the findings in mice are not relevant at all.

Page 21: the discussions on C. difficile are not relevant at all.

No data to support the statement on “vancomycin resistance genes were abundant in ruminal microbial genomes. Based on Table 1, expression of those genes were not detected in the metatranscriptome datasets (low RKPM values).

Page 22: second paragraph, no data to support these and not relevant

Reviewer #2 (Remarks to the Author):

Manuscript: Selection pressure and mobile genetic elements shape the antibiotic resistome in the rumen microbiome

This is an extensive piece of work, based on genome mining of the Hungate Project genomic resources, allowing analysis of hundreds of reference genomes of cultured ruminal bacteria. Indeed, there are few studies describing ARGs among particular ruminal bacteria. In fact, the authors should clarify that the major contribution of this manuscript is to associate ARM genes and bacterial species of the rumen. In this sense, the Title is “too general”, “already know”, and poorly descriptive of the work done. Something like: “Antibiotic resistance genes in the species of the rumen microbiota” will be clearer. What the authors consider “selection pressure” is very confuse along the paper, see for instance comments in page 18. In general, the manuscript contains many interesting features, but embedded in an unclear style of writing, with many redundancies and an oversized discussion, where very known facts are sometimes presented as something new. The authors should try to simplify and present clearly the message they want to convey.

Page 1. Abstract. "... but their prevalence in the gastrointestinal tract (GIT) of ruminants..." Revise the construction of this sentence. Prevalence of infections caused by multidrug....in the intestinal tract (?), What is not well understood? The prevalence of infections?

Page 2: "together with members of the family Enterobacteriaceae (especially *Klebsiella pneumoniae*)". Please include here *Escherichia coli*, by far more involved than *Klebsiella* in human AMR infections. For a reference:

Vila, J., Sáez-López, E., Johnson, J. R., Römling, U., Dobrindt, U., Cantón, R., ... & Bosch, J. (2016). *Escherichia coli*: an old friend with new tidings. *FEMS microbiology reviews*, 40(4), 437-463.

Page 2: "*Clostridium jejunii*" correct name is *Clostridium jejuni*.

Page 2: "...genetically diverse microbiota that live in close physical contact... Probably the authors wanted to say "genetically diverse microbiota where bacterial organisms live in close contact".

Page 3: "...the occurrence and distribution of ARGs among ruminal bacteria is not well documented and the potential mobility of the ARGs within the ruminal microbiota has not been investigated".

The authors should clarify that the major contribution of this manuscript is to associate ARM genes and bacterial species of the rumen.

For ARGs within the rumen "microbiota", a number of references are available, for instance, we have here two references:

Auffret MD, Dewhurst RJ, Duthie CA, Rooke JA, John Wallace R, Freeman TC, Stewart R, Watson M, Roehe R. The rumen microbiome as a reservoir of antimicrobial resistance and pathogenicity genes is directly affected by diet in beef cattle. *Microbiome*. 2017. ;5(1):159.

Thomas M, Webb M, Ghimire S, Blair A, Olson K, Fenske GJ, Fonder AT, Christopher-Hennings J, Brake D, Scaria J. Metagenomic characterization of the effect of feed additives on the gut microbiome and antibiotic resistome of feedlot cattle. . *Sci Rep*. 2017 ;7(1):12257

Page 4: Correct: "80.8% of the aminoglycosides" (aminoglycosides)

Page 7. Correct "with *Citrobater*", should be "*Citrobacter*".

Page 7. Should be: NLAE-zl-C500.... was the only species...

Page 7. Correct: "*Proteus mirabilis*"

Page 10. Of course, the fact that genomes with tet(W) also harbor genes involved in plasmid transfer do not necessarily mean that tet(W) is harbored in these plasmids.

Page 10. "no positive matches were obtained...". The conclusion "these mechanisms alone cannot explain the widespread dissemination of the tetracycline resistance" is uncertain, as we are unsure if the bioinformatic methodology was the appropriate. Probably other methods, as PlacNet (Vielva, L.,

de Toro, M., Lanza, V. F., & de la Cruz, F. (2017). PLACNETw: a web-based tool for plasmid reconstruction from bacterial genomes. *Bioinformatics*, 33(23), 3796-3798) should be more suitable.

Page 12. The purposes of the selection pressure analysis of tet resistant genes should be explained. A resistance gene can evolve (by non-synonymous substitutions) to reach higher resistance to the antibiotic. That is not probably the case here, as the tetracycline MIC generally provided by tet(X) is very high. However, as tet(X) encodes a tetracycline degradation protein, the MIC might not be so high (and therefore, evolvable) in the case in new organisms that have recently received the gene by HGT. Are the variant genes associated with different species?

Page 14. "In general, tetracycline resistance genes presented the highest level of expression while vancomycin resistance was the least expressed"

Is that true for tet(X)? Also for the other tet? If the expression is high, tetracycline degradation should probably be higher. (later: yes, that is visible in Table 1).

Page 17: Correct: "Clostridium and Lachnoclostridium", and later, "Selenomonas"

Page 18, line 6 "commensal"

A comparison with similar known ICEs should be included.

Page 18. "Positive selection pressure on tet(W)". The authors should make clear the distinction between "selection making more frequent tetX" in a particular species or in the microbiome, probably mediated by antibiotic pressure (but not altering the tetX gene as such), and "selection of the genetic variants of the tetX gene" (for instance increasing MICs), which is an evolution of the gene, as was analyzed in the manuscript looking at synonymous/non-synonymous changes in tetX.

Page 18. After reference 27, the authors could add the reference of a recent excellent review, Markley JL and Wencewicz TA (2018) Tetracycline-Inactivating Enzymes. *Front. Microbiol.* 9:1058. doi: 10.3389/fmicb.2018.01058.

Page 20. In addition, our results demonstrate a positive selective pressure on the tet(W) gene. For organisms containing tet(W)? Selection pressure to improve tet(W) evolution to better efficiency?

Page 20. "Betalactam resistance in the ruminal genomes". Many of these genes could be intrinsic resistance, not necessarily linked with antibiotic selection.

Page 22. First sentence. ..."as exemplified by the genome of E. coli PA-

3, which contains five resistance genes conferring resistance to four distinct antibiotics". What new? known for decades.

Page 21."that our in vitro assays confirmed only 31.6% of the resistance phenotype predicted in silico". Indeed that is of interest. The authors make the interpretation that our lab conditions do not mimic the rumen conditions. Not clear how that is deduced, as what they show in the paragraph is "different expressions in dairy and beef cattle and the human gut". The idea is that genes can be expressed in some animals, and not in others or in the lab?

Response to the Referees comments - manuscript NCOMMS-19-00128

Reviewer 1

The manuscript titled “Selection pressure and mobile genetic elements shape the antibiotic resistome in the rumen microbiome” applied bioinformatic based approaches to identify ARGs from the known microbial genomes and have applied some validation approaches to confirm them. Although the concept of identifying ARGs using different bioinformatic tools is not new and novel, the authors made good efforts with the aim to verify the presence of ARGs in the rumen microbial cultures and/or rumen microbiome. But there are some major weaknesses on the current version, which prevents its publication at NC.

1. Many conclusions drawn without proper data support. One of the issues is the lack of proper experimental design to generate those data:

i). The authors applied three in silico tools to identified ARGs from 435 genomes of rumen bacteria and archaea, and reported significant different results among these methods (Pages 4-6). It is very common that different bioinformatic tools usually generate varied outcomes for the same sequence dataset. Here, it is extremely important that authors should consider the “common findings” identified from all three methods to be likely the “true” ARGs. Based on Figure 7, the AR in vitro assay also showed similar trend with the ARG identified from all three methods.

Response: We thank the reviewer for recognizing the relevance of this work and for the helpful comments. In our experimental design, we carefully chose bioinformatic tools that used: 1) distinct ARG datasets (e.g. Resfinder – acquired resistance genes; Resfams – protein sequence database; ARG-ANNOT – existing and putative new ARGs) and 2) distinct search methods to detect putative antibiotic resistance genes (ARG-ANNOT and Resfinder – pairwise sequence alignment (BLAST); Resfams – hidden Markov models (HMMs)). Therefore, due to differences in sensitivity and specificity and also the distinct search strategies used by each software, some ARGs were detected by only one or two of these bioinformatic approaches. If only the “common findings” from all three methods were considered as “true” ARGs, the number of false negatives in the ruminal microbial genomes could be too high. This hypothesis is supported by data shown in Figure 7, which shows that some resistance phenotypes that were predicted by only one or two bioinformatic tools were experimentally confirmed in our *in vitro* assays. Since this study is also exploratory regarding the distribution of resistant genes in the species of rumen bacteria, we decided to consider as “true” ARGs all hits that were detected by at least two bioinformatic tools. The materials and methods section has been rewritten to clarify this point.

ii). The true and biological relevant functions of identified ARGs are not verified. In addition, the selection pressure and mobile genetic elements were only done based on “dry” analysis

and lack of biological relevance. Need validation assay or another approach to verify the results:

For example, comparison the rumen resistomes between cattle fed with antimicrobials vs without antimicrobials or fed with antimicrobials and then withdraw the antimicrobials to validate the selection pressure (either positive or negative) on rumen resistomes. Similarly, the mobile genetic elements should be verified in the true data set. In addition, the authors calculated dn/ds based on genomes. Can such value be validated in the in vitro assays?

Response: Our *in vitro* assays confirmed that some of the predicted ARGs confer resistance to the bacterial strains harboring them. Unfortunately, there is a limited number of ruminal strains readily available from culture collections to validate the biological function of all the predicted ARGs. Nonetheless, we were able to confirm experimentally the presence of the ICE_RbtetW_07 in *Blautia schinkii* 10518, which shows sequence conservation in different genomes of ruminal bacteria. Experiments to validate *in vitro* or *in vivo* the selective pressure imposed by antibiotic use in the evolution of antibiotic resistance genes would require thousands of generations, as demonstrated by Jochumsed et al (Nat. Commun. 7: 13002, 2016), which would not be feasible for this study. However, to strength our claims about selective pressure on the tetracycline resistance genes, we have confirmed our results using two additional bioinformatic tools (Tajima test and Fu and Li test). These results have now been added to the manuscript.

iii). The comparison between rumen microbiome and human fecal microbiome does not make any sense. The overall introduction, discussion and conclusions drawn on the linkage between rumen resistome and human AMRs is vague and lack of data support. Although it is trendy to make the linkages between livestock resistome with human AMR concerns, the conclusion can not be just drawn based on one beef cattle rumen metatranscriptome, one dairy cattle rumen metatranscriptome and one human fecal metatranscriptome. If authors aim to identify the “evidence” between rumen and human, samples should be collected from cattle rumen, fecal, soil and the fecal from the person who have contact with cattle. With only three random datasets, it does not provide data that support the conclusion.

Response: Thank you for your comment. Because validation of the linkages between livestock resistome with human AMR requires extensive work, beyond the scope of the present study, we decided to excluded all human metatranscriptomes from our analyses and focused on the resistome of the microbiota representing the core microbiome of the rumen. For this, we expanded the analyses of rumen metatranscriptomes to 15 datasets (five from beet cattle, five from dairy cattle and five from sheep) to strength our claims regarding the distribution and expression of antibiotic resistance genes in the rumen ecosystem (Figure 8). The manuscript has been revised accordingly.

2. *Need more stringent cut-offs and more datasets. It is not clear how the cut offs (70% identify and 60% coverage) were determined (pages 24 and 25). How do authors know these are strong enough?*

Response: We thank the reviewer for this comment. The cut-off values used in our analysis are in agreement with previous studies that aimed to identify orthologs in genomic sequences (Scientific Reports 6: 37811, 2016; PlosOne 9: e92798, 2014; Int. J. Syst. Evol. Microbiol. 66: 1100-1103, 2016). In addition, the domain region of many ARGs are much conserved. Although we could have used more stringent cut-off values in our analyses, we decided to use the values reported in the manuscript for two main reasons: 1) three bioinformatic approaches were used to predict the ARGs and only hits identified by at least two algorithms were regarded as “true”; 2) we wanted to detect both existing and potentially new antibiotic resistance genes in the microbial genomes. Moreover, it should be pointed out that for the majority of the genes detected using Resfinder, the percentage of both identity and coverage was always higher than 90%. Regarding the datasets, we have now expanded our rumen metatranscriptomes analyses to 15 datasets, including beef and dairy cattle and sheep, as mentioned above.

What is the cut-off for RKPM for the gene expression? Based on Table 1, the RKPM values are low (0.01-0.1) (6 out of 8 ARGs) for beef cattle rumen metatranscriptome which should be considered as undetectable. For gene expression, the RKPM should be higher than 0.2 or 0.3 to be considered as expressed. Similar, (6 out of 10) in dairy rumen metatranscriptome should be also not expressed.

Response: Thank you for your comment. Because RPKM cut-offs are not well defined in the literature, we initially did not establish a cut-off value for these analyses. However, following the reviewer’s suggestion and based on the work of Warden et al. (Int. J. Comput. Bioinfo. In Silico Model, 2: 285-292, 2013), we reanalyzed our gene expression data and only genes with RKPM higher than 0.3 were considered as expressed. The manuscript has been revised to include these data and the cut-off value has been reported in the Materials and Methods section.

3. *The focus on tetracycline resistance is more computer bases, not biology driven.*

i). tet(w) is identified in a couple of Bifidobacterial species, what does this mean?

Response: Regarding the tetracycline resistance, we have now experimentally confirmed the presence of an ICE carrying the tetracycline genes in the genome of a ruminal bacteria. Bifidobacteria are abundant members of the infant gut microbiota, being also distributed in the gut microbiome of young livestock. Due to its potential beneficial role in the intestinal tract and health promotion, several strains have been used as probiotics for humans and animals. Therefore, strains carrying antibiotic resistance genes are highly undesirable in this bacterial group and may have negative consequences to human health, through animal products such as milk and meat, especially in infants, in which the gut microbiota is more susceptible to perturbations and long-term disturbances compared to adult individuals.

Although this is an interesting observation, we do not believe that these results should be emphasized in our discussion session.

ii). Also, the authors need to clearly indicate how and when the tetracycline has been admitted in the ruminant production and how the identified tet(s) are relevant to the ruminant production system.

Response: We have modified the manuscript to accommodate the reviewer's concern and information about how and when tetracycline was introduced in livestock production has been included in the discussion section.

iii). Again, the linkage to human is very weak.

Response: The manuscript has been extensively revised and any linkages between livestock resistome and human AMR have been toned down or completely removed from the text. The focus was kept primarily on the rumen resistome.

Other comments:

Through the manuscript, it should be ruminal microbial genomes, not ruminal genomes. One refers to microbes, the other refers to host genome.

Response: We thank the reviewer for calling our attention to this point. The text has been changed as indicated by the reviewer.

Suggest moving the validation using metatranscriptomes section to be the end of results to make the flow better.

Response: The data on metatranscriptomic analyses have been moved to the end of Results section as requested by the reviewer and the Material and Methods section has been revised accordingly.

Page 2: the first paragraph is not very relevant, suggest to simplify

Response: The first paragraph has been changed as suggested by the reviewer.

Second paragraph, please make all statements to be more specific especially about the AM use in livestock animals. Here should include information on ruminants and limit the cite the information from swine and poultry production.

Response: The text has been changed as suggested by the reviewer. The paragraph is focused primarily on ruminant production.

Pages 4-6: very confusing. It would be better if authors can summarize the outcomes from three approaches using a Table, since the outcome of the comparison among three methods is one of the conclusions.

Response: Although some of the results have been summarized in Figures 1 and 2 of the Supplementary material, we agree with the reviewer that the outcomes of the bioinformatics

approaches can be made more evident to the reader if presented as a Table. We have now included a new Table 1 in the manuscript to summarize these outcomes as suggested by the reviewer.

Page 7 and page 22: the ARG from E. coli PA-3: It has been well known varied E. coli strains carry ARGs. The authors should here indicate how this strain differ from other strains? Also, does this train specific to rumen? Could this be an environmental organism?

Response: We agree with the reviewer that the presence of ARGs in *E. coli* is not new. However, *E. coli* PA-3 was isolated from the rumen and reports about ruminal strains of *E. coli* harboring multiple ARGs are very limited (including the *E. coli* PA-3 strain). Strains of *E. coli* (and other proteobacteria) can be found in the rumen of young and adult ruminants, indicating that this organism is also a member (albeit less abundant) of the rumen microbiome. We have modified the manuscript to indicate more clearly that *E. coli* PA-3 is a ruminal strain.

Page 9: not clear what is the purpose to draw the network and what do these networks suggest?

Response: Figure 3 has been removed from the manuscript.

Suggest to significantly reduce the discussions on human microbiome, really focused on rumen microbiome and maybe ruminant cattle microbiome.

Response: The manuscript has been modified accordingly to accommodate the reviewer suggestion. Discussion is focused on rumen microbiome.

Page 19: statement (tet(w) gene in butyrvibrio fibrisolvens) based on citation no. 30. No such resistance from this manuscript (figure 7). What does this suggest?

Response: It should be noted that the *Butyrvibrio fibrisolvens* strains analyzed in our work are distinct from the one reported in citation no. 30 (*B. fibrisolvens* 1230). Also, our data show that the tet(W) gene is not widely distributed among the genomes of *B. fibrisolvens* isolated from the rumen. Our analysis predicted the tet(W) gene in only one *B. fibrisolvens* genome (strain AR40) out of nine *B. fibrisolvens* genomes analyzed using different bioinformatic tools.

Page 20: more data are needed to support the conclusions drawn in the first paragraph.

Response: We have now experimentally confirmed the presence of the ICE carrying tetracycline gene in *Blautia schinkii* DSM 10518 using PCR and the genetic organization of the ICE was confirmed by shotgun sequencing. These results provides biological evidence for the presence of these genetic elements in ruminal bacteria and strength the idea that they could be transferred to other bacteria within the GIT of ruminants. However, experimental

confirmation that these genetic elements can be transferred to other commensal or pathogenic bacteria will require more extensive work, which beyond the scope of the present study.

The first part of 2nd paragraph is also vague and not relevant to the current study, and the findings in mice are not relevant at all.

Response: The introduction section has been extensively revised and these paragraphs have been changed following the reviewers suggestions. Thank you.

Page 21: the discussions on C. difficile are not relevant at all.

Response: The text has been changed and discussions on *C. difficile* have been removed as suggested by the reviewer.

No data to support the statement on “vancomycin resistance genes were abundant in ruminal microbial genomes. Based on Table 1, expression of those genes were not detected in the metatranscriptome datasets (low RKPM values).

Response: We thank the reviewer for this observation. We expanded our metatranscriptomic analyses to 15 rumen datasets from beef/dairy cattle and sheep. These results confirmed that vancomycin resistance genes have low expression in the rumen microbiome. We have changed the discussion accordingly, based on these new data.

Page 22: second paragraph, no data to support these and not relevant

Response: This paragraph has been extensively revised and most inferences about potential linkages between AMR in ruminants and humans have been removed. Discussion of antibiotic resistance is centered in the rumen microbiota and based on the analyses of a larger number of ruminal metatranscriptome datasets.

Reviewer 2

Manuscript: Selection pressure and mobile genetic elements shape the antibiotic resistome in the rumen microbiome

This is an extensive piece of work, based on genome mining of the Hungate Project genomic resources, allowing analysis of hundreds of reference genomes of cultured ruminal bacteria. Indeed, there are few studies describing ARGs among particular ruminal bacteria. In fact, the authors should clarify that the major contribution of this manuscript is to associate ARM genes and bacterial species of the rumen. In this sense, the Title is “too general”, “already know”, and poorly descriptive of the work done. Something like: “Antibiotic resistance genes in the species of the rumen microbiota” will be clearer. What the authors consider “selection pressure” is very confuse along the paper, see for instance comments in page 18. In general, the manuscript contains many interesting features, but embedded in an unclear style of

writing, with many redundancies and an oversized discussion, where very known facts are sometimes presented as something new. The authors should try to simplify and present clearly the message they want to convey.

Response: We thank the reviewer for the positive overview about this study and for the valuable comments that helped to improve the manuscript. The title suggestion has been accepted and incorporated in the revised manuscript; the introduction and discussion have been modified to clarify the association between AMR genes and bacterial species of the rumen. In addition, discussion has been shortened and we hope that the revised manuscript demonstrates the main message with greater clarity.

Page 1. Abstract. "... but their prevalence in the gastrointestinal tract (GIT) of ruminants..."
Revise the construction of this sentence. Prevalence of infections caused by multidrug....in the intestinal tract (?), What is not well understood? The prevalence of infections?

Response: We agree with the reviewer in this point. The abstract has been rewritten and this sentence has been clarified. Thank you.

Page 2: "together with members of the family Enterobacteriaceae (especially Klebsiella pneumoniae)". Please include here Escherichia coli, by far more involved than Klebsiella in human AMR infections. For a reference:

Vila, J., Sáez-López, E., Johnson, J. R., Römling, U., Dobrindt, U., Cantón, R., ... & Bosch, J. (2016). Escherichia coli: an old friend with new tidings. FEMS microbiology reviews, 40(4), 437-463.

Response: We agree with the reviewer about the relevance of *E. coli* in human AMR infections. However, following a suggestion made by Reviewer 1 to simplify the Introduction section, we decided to remove these sentences.

Page 2: "Clostridium jejunii" correct name is Clostridium jejuni.

Response: This has been corrected as indicated. Thank you.

Page 2: "...genetically diverse microbiota that live in close physical contact... Probably the authors wanted to say "genetically diverse microbiota where bacterial organisms live in close contact".

Response: The text has been changed as indicated by the reviewer. Thank you.

Page 3: "...the occurrence and distribution of ARGs among ruminal bacteria is not well documented and the potential mobility of the ARGs within the ruminal microbiota has not been investigated". The authors should clarify that the major contribution of this manuscript is to associate ARM genes and bacterial species of the rumen.

For ARGs within the rumen "microbiota", a number of references are available, for instance, we have here two references:

Auffret MD, Dewhurst RJ, Duthie CA, Rooke JA, John Wallace R, Freeman TC, Stewart R, Watson M, Roehe R. *The rumen microbiome as a reservoir of antimicrobial resistance and pathogenicity genes is directly affected by diet in beef cattle. Microbiome. 2017. ;5(1):159.*
Thomas M, Webb M, Ghimire S, Blair A, Olson K, Fenske GJ, Fonder AT, Christopher-Hennings J, Brake D, Scaria J. *Metagenomic characterization of the effect of feed additives on the gut microbiome and antibiotic resistome of feedlot cattle. . Sci Rep. 2017 ;7(1):12257*
Response: We agree with the reviewer and thank you for this comment. This sentence has been modified to accommodate the reviewer's concerns and to emphasize the contribution of this work associating the ARGs with bacterial species of the rumen.

Page 4: Correct: "80.8% of the aminoglycosides" (aminoglycosides)

Response: The text has been changed as indicated by the reviewer. Thank you.

Page 7. Correct "with Citrobater", should be "Citrobacter".

Response: The text has been changed as indicated by the reviewer. Thank you.

Page 7. Should be: *NLAE-zl-C500.... was the only species...*

Response: The text has been changed as indicated by the reviewer. Thank you.

Page 7. Correct: "Proteus mirabilis"

Response: The text has been changed as indicated by the reviewer. Thank you.

Page 10. *Of course, the fact that genomes with tet(W) also harbor genes involved in plasmid transfer do not necessarily mean that tet(W) is harbored in these plasmids.*

Response: We thank the reviewer for this observation. The text has been changed accordingly to clarify this point.

Page 10. *"no positive matches were obtained...". The conclusion "these mechanisms alone cannot explain the widespread dissemination of the tetracycline resistance" is uncertain, as we are unsure if the bioinformatic methodology was the appropriate. Probably other methods, as PlacNet (Vielva, L., de Toro, M., Lanza, V. F., & de la Cruz, F. (2017). PLACNETw: a web-based tool for plasmid reconstruction from bacterial genomes. Bioinformatics, 33(23), 3796-3798) should be more suitable.*

Response: We understand the concern of the reviewer. Our main interest was to evaluate if the *tet(W)* genes were located near or within mobile genetic elements. Although we were able to find several homologs protein associated with the bacterial conjugation machinery using different computational tools, the presence of complete genetic elements showing the typical structural organization of a plasmid were not verified in these genomes. It should also be emphasized that PlasmidFinder has been successfully used to identify many plasmids in bacterial genomes, showing that this tool is also a robust approach. Although PLACNETw is another interesting tool, it is restrictive regarding the input data file format (.gz), which

limited the analyses of the rumen microbial genomes. These sentences have been modified to clarify this point.

Page 12. The purposes of the selection pressure analysis of tet resistant genes should be explained. A resistance gene can evolve (by non-synonymous substitutions) to reach higher resistance to the antibiotic. That is not probably the case here, as the tetracycline MIC generally provided by tet(X) is very high. However, as tet(X) encodes a tetracycline degradation protein, the MIC might not be so high (and therefore, evolvable) in the case in new organisms that have recently received the gene by HGT. Are the variant genes associated with different species?

Response: The purposes of the selection pressure analysis of the tet resistant genes has been explained in the Material and Methods section as requested by the reviewer. We agree with the comments made for the tet(X) gene, but it should be noted that we did not identified this gene in the genomes of ruminal bacteria, therefore we cannot speculate if it is evolvable or not. In the case of tet(W), the variants were indeed associated with different species of bacteria, and we believe this is clearly shown in our results.

Page 14. "In general, tetracycline resistance genes presented the highest level of expression while vancomycin resistance was the least expressed"

Is that true for tet(X)? Also for the other tet? If the expression is high, tetracycline degradation should probably be higher. (later: yes, that is visible in Table 1).

Response: Because we did not detect the tet(X) gene in rumen microbial genomes, we cannot comment on its expression in the ruminal metatranscriptomes. Regarding the other tet genes, the expression was higher for tet(O), tet(Q), tet(W) and tet(37), compared to the other ARGs, as shown by Figure 8.

Page 17: Correct: "Clostridium and Lachnospirillum", and later, "Selenomonas"

Response: The text has been corrected as suggested by the reviewer. Thank you.

Page 18, line 6 "commensal"

A: The text has been corrected as suggested by the reviewer. Thank you.

A comparison with similar known ICEs should be included.

Response: We understand the reviewer concern, but the ICE_RbtetW_07 identified in this study shows little homology to other ICEs previously reported, which makes direct comparison of the genetic organization of ICE_RbtetW_07 with other conjugative elements very difficult. However, a new supplementary Figure 8 showing the phylogenetic relationship of ICE_RbtetW_07 with other ICEs has been included in the manuscript.

Page 18. "Positive selection pressure on tet(W)". The authors should make clear the distinction between "selection making more frequent tetX" in a particular species or in the

microbiome, probably mediated by antibiotic pressure (but not altering the tetX gene as such), and "selection of the genetic variants of the tetX gene" (for instance increasing MICs), which is an evolution of the gene, as was analyzed in the manuscript looking at synonymous/non-synonymous changes in tetX.

Response: We believe that the reviewer is referring to the analysis of *tet(W)* in this query. We have modified these sentences to clarify the idea that positive selection refers to the evolution of the gene and not to its frequency in the population because of antibiotic pressure. Thank you.

Page 18. After reference 27, the authors could add the reference of a recent excellent review, Markley JL and Wencewicz TA (2018) Tetracycline-Inactivating Enzymes. Front. Microbiol. 9:1058. doi: 10.3389/fmicb.2018.01058.

Response: The citation indicated by the reviewer has been referenced in the manuscript. Thank you.

Page 20. In addition, our results demonstrate a positive selective pressure on the tet(W) gene. For organisms containing tet(W)? Selection pressure to improve tet(W) evolution to better efficiency?

Response: The reviewer is correct. The selection pressure analysis was carried out in the genomes containing the *tet(W)* gene. As explained above, the positive selection refers to the evolution of the gene and not to its frequency in the population because of antibiotic pressure. This sentence has been modified to clarify this idea. Thank you.

Page 20. "Beta-lactam resistance in the ruminal genomes". Many of these genes could be intrinsic resistance, not necessarily linked with antibiotic selection.

Response: We agree with the reviewer that some of the beta-lactam resistance genes could be related with intrinsic resistance, but we cannot discard the hypothesis that some of these genes were acquired by horizontal transfer. The sentence has been rewritten to clarify this idea.

Page 22. First sentence. ..."as exemplified by the genome of E. coli PA-3, which contains five resistance genes conferring resistance to four distinct antibiotics". What new? known for decades.

Response: As explained above to Reviewer 1, we agree with the reviewers that the presence of ARGs in *E. coli* is not new. However, *E. coli* PA-3 was isolated from the rumen and reports about ruminal strains of *E. coli* harboring multiple ARGs are very limited (including for the *E. coli* PA-3 strain). *E. coli* (and other *Proteobacteria*) can be found in the rumen of young and adult ruminants, indicating that this organism is also a member (albeit less abundant) of the rumen microbiome.

Page 21."that our *in vitro* assays confirmed only 31.6% of the resistance phenotype predicted *in silico*". Indeed that is of interest. The authors make the interpretation that our lab conditions do not mimic the rumen conditions. Not clear how that is deduced, as what they show in the paragraph is "different expressions in dairy and beef cattle and the human gut". The idea is that genes can be expressed in some animals, and not in others or in the lab?

Response: We have modified the discussion section and comparisons between human and ruminant datasets are no longer valid. However, the point we wanted to make in these sentences is that an AMR gene might be present in the genome, but not necessarily functional or expressed under the standard laboratory conditions established for antibiotic resistance testing. In addition, although the *in vitro* analyses were subjected to the growth conditions of individual bacterial strains, the expression of ARGs in rumen metatranscriptomes reflect more closely the *in vivo* conditions of the rumen ecosystem. Differences in gene expression in the metatranscriptomes have also been clarified in the discussion.

Reviewers' comments:

Reviewer #1 (Remarks to the Author):

The authors also have addressed the reviewers' comments promptly. The revised version has improved significantly with way better flow and more detailed information. Suggest major revision of the revised version due to the following issues:

Suggest to delete all abbreviations in abstract except tet(w), also define ICE in the introduction not discussion section.

Page 2, the statement "The rumen ecosystem is colonized..... which facilitates the exchange of genetic material between members of the microbiota and organisms that transit in the ruminant gastrointestinal tract (GIT)"

needs some clarification about organisms that transit in the ruminant gastrointestinal tract (GIT). Do you mean that the transferred the genetic elements could transit through the whole gut? If so, what is the issue. Based on the papers the authors cited, the genetic material transfer and/or exchange is mainly referred to the rumen microbes, not clear whether it actually pass through the whole gut. The authors need to find proper citations to support this statement.

Page 2: the statement "...and are raised in close proximity with humans.". This is not relevant at all. Does this suggest the potential increased prevalence of ARGs transfer between human and cattle? Also, is there any evidence that this is happening or may happen? Suggest to delete this part.

Page 3: the authors performed analysis on 435 genomes of bacteria and archaea, but why the main findings only include bacteria, not archaea. The ARG genes were detected as shown in Table 1.

Page 5: "as in all strains of the genus Enterococcus and Staphylococcus". Should be all species of the genus....or all strains of particular species

Page 5: "No antibiotic resistance was detected in genomes belonging to the Spirochaetes, Fibrobacteres and Fusobacteria phyla, but it should be noted that the number of genomes representing these taxa were much lower compared to the other phyla analyzed in this study."

Do you mean no antibiotic resistance genes? Also, the Resfams did detect 27, 4 and 2 ARGs of these phyla respectively as reported in Table 1. Were they just noise? If so, it should be presented as ND in

Table 1 for these phyla. Please also include the archaea results here, indicating the low prevalence (or not detected) in the 10 archaeal genomes.

Page 6: “strains of *Staphylococcus epidermidis*, respectively”. How many strains? Better to specify.

Page 6: “Beta-lactam resistance genes were concentrated in the family Enterobacteriaceae, in the genus *Bacteroides* and in two clades harboring the genera *Selenomonas*, *Staphylococcus*, *Succinoclasticum* and *Bacillus*.”. There is no evidence of the colonization of *Bacillus* in the rumen. Although they are included in the rumen microbial genomes, this genus should be associated with feed, Suggest authors to have some discussions on the including this genus for data analysis and results.

Page 8: “From these, the protein under the accession number D1PK82 in Uniprot database, was the most prevalent and showed 98.9% to 100% sequence identity across the genomes of ruminal bacteria.”. Should it be the genes encoding the protein D1PK82 was the most prevalent....? Similar, through this paragraph, it should be genes encoding proteinsXX in the flanking regions.

Page 9: should be “ICE_RbtetW_07 had low sequence homology...”. Please define which seven genomes either in text or legend of Figure 4. Although they are shown in the figure, it is extremely difficult to tell, can them be labeled them differently?

Page 10: should be “a fragment of 4794 bp covering the region from the *VirD4* to...”. Here refers gene not protein, so the gene name should be italic.

Page 11: Any reason why the size of PCR products differ between two batches (Lane 1 and Lane 2)? Suggest to include Fig 5 as supplementary not the main results as this figure does not look good. If the authors insist, please include a better photo as well as to include negative control, and possibly, positive control.

Page 11: For this purpose, the number of...

Page 15: what do the numbers in Table 2 mean? No explanation in table footnotes. Also, it seems there are some overlapping between Table 2 and Figure 7, maybe one of them better be supplementary?

Page 17: suggest to revise the figure 8 to include only expressed data in the figure and state that most of detected genes were not expressed in the text. Also, maybe highlight Beef, Dairy cattle and Sheep on top of figure. Assume the authors obtained 5 metatranscriptomes each, but not sure. Also, better to describe a bit about the nature of these 15 metatranscriptome datasets. Were they generated under “antimicrobial” selective pressure or not? Could authors also determine which could be the host organisms express these genes in Beef/Dairy Cattle and sheep? Or at least show the abundance of the organisms that carry these genes in the metatranscriptome datasets?

The discussion is kind of weak, and suggest to move some of the discussions in the results section to strengthen the discussion section.

Page 19-20: “Our in silico results demonstrated....as exemplified by the genome of E. coli PA-3, a ruminal isolate that contains five genes conferring resistance to four distinct antibiotics?”

So what do your these suggest? Is E. coli PA-3 a problem? It would be better if authors expand their discussions here to look at the prevalence of active Pro bacteria Phylum in the 15 metatranscriptome datasets rather than focus on E. coli PA-3. The overall E. coli population in the rumen is low and the role of this strain should be not important to the rumen.

Page 20: “ In addition, our transcriptomic analyses showed higher expression of tetracycline resistance genes..... which also agreed with our in vitro experiments with pure cultures of ruminal bacteria”.

Again, what do these suggest? Please indicate the nature of the metatranscriptome dataset and indicate future survey of expression of detected ARGs detected in this study under different feeding, environment, through different growth stage and so on.

Also, the authors should also point out to survey these ARGs through GIT of cattle to indicate their potential transit through the gut by ICEs.

Reviewer #2 (Remarks to the Author):

The manuscript has been modified in a satisfactory way, and there is no more queries from my part.

Consider these minor changes:

Table 1: Please correct "Fusic acid". Should be "Fusidic acid"; "Oxazolidone" should be "Oxazolidinone", "Fenicol" should be "Phenicol". Better in plural: aminoglycosides, beta-lactams, sulphonamides, tetracyclines. Spell MLS: Macrolides, Lincosamides, Streptogramin B

Please apply these changes to the antibiotic designations at the end of page 21.

Page 6, fifth line from the bottom. Should be: "Bacteroides ovatus... was the only species in Bacteroidetes possessing genes for sulphonamide resistance". Check if that is right as I'm writing it.

Two lines below: "Proteus mirabilis"

Reviewers' comments:

Reviewer #1 (Remarks to the Author):

The authors also have addressed the reviewers' comments promptly. The revised version has improved significantly with way better flow and more detailed information.

Response: We very much appreciate the reviewer's detailed evaluations and suggestions. The manuscript has been revised thoroughly according to the reviewer's advice.

Suggest major revision of the revised version due to the following issues:

Suggest to delete all abbreviations in abstract except tet(w), also define ICE in the introduction not discussion section.

Response: All abbreviations in the abstract have been excluded as suggested by the reviewer and the term ICE is now defined in the introduction as suggested by the reviewer.

Page 2, the statement "The rumen ecosystem is colonized..... which facilitates the exchange of genetic material between members of the microbiota and organisms that transit in the ruminant gastrointestinal tract (GIT)" needs some clarification about organisms that transit in the ruminant gastrointestinal tract (GIT). Do you mean that the transferred the genetic elements could transit through the whole gut? If so, what is the issue. Based on the papers the authors cited, the genetic material transfer and/or exchange is mainly referred to the rumen microbes, not clear whether it actually pass through the whole gut. The authors need to find proper citations to support this statement.

Response: We agree with the reviewer that this sentence needs some clarification. The text has been revised to describe more precisely how the mechanisms promoting exchange of genetic material between microbial species could affect the rumen microbiota and other individuals/species passing through the ruminant gastrointestinal tract. We also added more specific citations to support these statements. Thank you.

Page 2: the statement "...and are raised in close proximity with humans.". This is not relevant at all. Does this suggest the potential increased prevalence of ARGs transfer between human and cattle? Also, is there any evidence that this is happening or may happen? Suggest to delete this part.

Response: We thank the reviewer for this suggestion. However, we believe that this statement is relevant to the ARGs context in ruminants, since livestock can shed ARGs in their feces and there is a risk of direct/indirect transfer of bacteria carrying ARGs to humans (e.g. farm workers, family members). Previous reports indicated an association between ARGs found in bacteria isolated from animals (including ruminants) and farm workers (Front. Microbiol. 8:818, 2017; J. Dairy Sci. 99(6): 4251-4258, 2016; Clin Microbiol Rev. 24(4):718-33, 2011). Moreover, a recently study showed that pig movements in the field contribute to the spread of a methicillin *Staphylococcus aureus* strain in a pig production system (mBio 9(6): e02142-18, 2018), and similar microbial dissemination may apply for cattle herds. Besides, mobile genetic elements (MGEs), such as class I integrin and plasmids, carrying ARGs with identical gene cassettes have been identified in *E. coli* and/or *Salmonella* isolated from both food-producing animals and humans (Clin. Infect. Dis. 49: 365-371, 2009; Lancet. Infect. Dis. 16:161-168, 2016), demonstrating the role of MGEs mediating the transfer of ARGs from animals to humans. Therefore, we are confident that there is enough support from the literature to maintain the statement in this paragraph.

Page 3: the authors performed analysis on 435 genomes of bacteria and archaea, but why the main findings only include bacteria, not archaea. The ARG genes were detected as shown in Table 1.

Response: We thank the author for pointing this out. No antibiotic resistance genes were identified in archaea using Resfinder and ARG-ANNOT. Resfams was the only computational tool that predicted ARGs in archaea, but it should be noted that these ARGs represent ABC efflux pumps, a mechanism that may not be related to antibiotic resistance. Since the predicted resistance phenotype could not be confirmed by other computational tools and only a low number of genomes were available for ruminal archaea (n=10), this group was not considered as carrying resistance genes in this study. Archaea are often intrinsically resistant to several antibiotics that inhibit bacteria in large part due to the fact that archaeal cells lack the specific targets where the drugs bind. We have changed the Results section to state that ARGs were not found in the Euryarchaeota phyla using Resfinder and ARG-ANNOT.

Page 5: "as in all strains of the genus Enterococcus and Staphylococcus". Should be all species of the genus....or all strains of particular species

Response: Thank you for this observation. The ARGs were identified in all species of the genus *Enterococcus*, including strains belonging to the same species, and among strains of *Staphylococcus epidermidis*. The text was modified accordingly to clarify this point.

Page 5: "No antibiotic resistance was detected in genomes belonging to the Spirochaetes, Fibrobacteres and Fusobacteria phyla, but it should be noted that the number of genomes representing these taxa were much lower compared to the other phyla analyzed in this study."

Do you mean no antibiotic resistance genes? Also, the Resfams did detect 27, 4 and 2 ARGs of these phyla respectively as reported in Table 1. Were they just noise? If so, it should be presented as ND in Table 1 for these phyla. Please also include the archaea results here, indicating the low prevalence (or not detected) in the 10 archaeal genomes.

Response: As we explained above for the Archaea phyla, Resfams can detect several ABC efflux pumps genes. The ABC efflux pumps represent one of the largest protein families in microorganisms, contributing not only to reduce the intracellular concentrations of toxic compounds, but also to the influx of substrates and other nutrients. Therefore, as stated in the manuscript, only ARGs that were identified by at least two bioinformatics tools applied in this study were considered for attributing a resistance phenotype to a particular species/strain. The Results section has been modified and this information is now clearly presented in the manuscript. We also included the *Euryarchaeota* results here, as requested by the reviewer. Thank you.

Page 6: "strains of Staphylococcus epidermidis, respectively". How many strains? Better to specify.

Response: The reviewer is correct. The text has been modified and the number of bacterial strains (two strains) was specified in the text as suggested. Thank you.

Page 6: "Beta-lactam resistance genes were concentrated in the family Enterobacteriaceae, in the genus Bacteroides and in two clades harboring the genera Selenomonas, Staphylococcus, Succiniclaticum and Bacillus". There is no evidence of the colonization of Bacillus in the rumen. Although they are included in the rumen microbial genomes, this genus should be associated with feed, Suggest authors to have some discussions on the including this genus for data analysis and results.

Response: Although we understand the reviewer concern, the assumption made by the reviewer is not accurate. Some strains of *Bacillus* have been previously isolated from the rumen of cattle and buffalo (Asian-Aust. J. Anim. Sci. (2013) 26: 50-58; Journal of Applied Microbiology (2001) 91: 636–64), and unpublished work from our research group also indicated the presence of several spore-forming bacilli in samples from the bovine rumen as well as in fecal samples. Although allochthonous strains could reach the rumen through feeds and water and persist in the forestomach or in the lower gastrointestinal tract (cecum and colon), we have no solid basis to discard these strains from our analysis. The same concept applies to other bacterial genera analyzed in this study, such as *Escherichia* and *Staphylococcus*, which are also found along the gastrointestinal tract of ruminants, including the rumen. The fact that the Hungate1000 Project has a low number of genomes belonging to these genera may reflect their lower abundance in the rumen ecosystem, compared to other taxa of the dominant anaerobic microbiota.

Page 8: *“From these, the protein under the accession number D1PK82 in Uniprot database, was the most prevalent and showed 98.9% to 100% sequence identity across the genomes of ruminal bacteria.”. Should it be the genes encoding the protein D1PK82 was the most prevalent....? Similar, through this paragraph, it should be genes encoding proteinsXX in the flanking regions.*

Response: We thank the reviewer for this observation. This information has now been added in the revised manuscript.

Page 9: *should be “ICE_RbtetW_07 had low sequence homology...”. Please define which seven genomes either in text or legend of Figure 4. Although they are shown in the figure, it is extremely difficult to tell, can them be labeled them differently?*

Response: The text has been modified as suggested, and the name of the seven genomes has been added in the legend of Figure 4. Unfortunately, we cannot change the MAUVE figure label because this is the default export file format.

Page 10: *should be “a fragment of 4794 bp covering the region from the VirD4 to...”. Here refers gene not protein, so the gene name should be italic.*

Response: The reviewer is correct. The text has been corrected as suggested. Thank you.

Page 11: *Any reason why the size of PCR products differ between two batches (Lane 1 and Lane 2)? Suggest to include Fig 5 as supplementary not the main results as this figure does not look good. If the authors insist, please include a better photo as well as to include negative control, and possibly, positive control.*

Response: We have repeated the DNA extractions and PCR amplifications and a better image has now been added to the revised manuscript. However, following the reviewer suggestion, we decided to include Figure 5 (now labeled as Figure 8) in the Supplementary material. Because ICE_RbtetW_07 represents a novel modular mobile genetic element, an appropriate positive control is not possible. However, we have included a negative control to the gel as requested.

Page 11: *For this purpose, the number of...*

Response: The sentence has been changed as indicated. Thank you.

Page 15: *what do the numbers in Table 2 mean? No explanation in table footnotes. Also, it seems there are some overlapping between Table 2 and Figure 7, maybe one of them better be supplementary?*

Response: We thank the reviewer for this observation. Table footnotes have been revised and this information is now clearly presented in the revised manuscript. We agree that Table 2 could

be moved to the Supplementary materials (now labeled as Table 1 in the Supplementary material). This has been changed accordingly, as suggested by the reviewer.

Page 17: suggest to revise the figure 8 to include only expressed data in the figure and state that most of detected genes were not expressed in the text. Also, maybe highlight Beef, Dairy cattle and Sheep on top of figure. Assume the authors obtained 5 metatranscriptomes each, but not sure. Also, better to describe a bit about the nature of these 15 metatranscriptome datasets. Were they generated under "antimicrobial" selective pressure or not? Could authors also determine which could be the host organisms express these genes in Beef/Dairy Cattle and sheep? Or at least show the abundance of the organisms that carry these genes in the metatranscriptome datasets?

Response: Figure 8 (now labeled as Figure 7 in the revised manuscript) has been changed as suggested by the reviewer and the revised manuscript shows only the ARGs that were expressed in a heatmap format. The origin of each transcriptome dataset (beef, dairy cattle or sheep) has been indicated at the bottom of the figure. Unfortunately, there is limited information about the nature of these metatranscriptomes, regarding the use of antibiotics to raise the animals that generated these datasets. Therefore, we cannot make assumptions about antimicrobial selective pressure in these animals. Because the sequences of the genes reported in our transcriptomes are highly conserved, it is not possible to determine the host organisms which they belong to. Similarly, to evaluate species abundance, the target genes should be constitutive and present in single copies in the host genomes, which is not possible to determine here. If we were working with metagenomes and metagenome-assembled genomes (MAGs), then it would be possible to evaluate the distribution of ARGs and abundance of host organisms as suggested by the reviewer.

The discussion is kind of weak, and suggest to move some of the discussions in the results section to strengthen the discussion section.

Response: Although we have carefully considered the reviewer suggestion, it is not clear which specific parts of the results section the reviewer would like to see in the discussion. We have modified some parts of the discussion to incorporate suggestions made by the reviewers and we hope that the revised manuscript meet the reviewer concerns.

Page 19-20: "Our in silico results demonstrated....as exemplified by the genome of E. coli PA-3, a ruminal isolate that contains five genes conferring resistance to four distinct antibiotics?"

So what do your these suggest? Is E. coli PA-3 a problem? It would be better if authors expand their discussions here to look at the prevalence of active Probacteria Phylum in the 15 metatranscriptome datasets rather than focus on E. coli PA-3. The overall E. coli population in the rumen is low and the role of this strain should be not important to the rumen.

Response: Although we agree with the reviewer that a single *E. coli* strain may not be so important for ARG dissemination in the rumen, it should also be pointed out that cattle are considered a natural reservoir for pathogenic strains of *E. coli*, including *E. coli* O157:H7 (FEMS Microbiology Letters (1993) 114: 79-84; Science, (1998) 281:1666-1668; Appl. Environ. Microbiol. (2004) 70: 5336–5342; Dairy Sci. (2011) 94:351–360). Generic *Escherichia coli* populations are often not so abundant in the rumen of cattle (less than 10^6 cells/ml), but management and nutritional practices, such as dietary switches from hay to grain-based ration, can cause significant increases in ruminal and fecal populations of *E. coli* and other taxa within the *Proteobacteria* phylum (J. Dairy Sci. (2003) 86:852–860; Microbiome (2017) 5:159). Therefore, these practices increase the abundance and diversity of ARGs in the rumen and could

increase the risk of spreading resistance genes between microbial species occupying the same ecological niche. The *E. coli* PA-3 strain was cited in our study as an example of ruminal *Enterobacteriaceae* carrying ARGs, but the discussion section has been modified to indicate more clearly that cattle can act as a natural reservoir for *E. coli* and that changes in *Proteobacteria* ratio, through dietary practices, may affect the abundance of ARGs in the rumen of cattle.

Page 20: *" In addition, our transcriptomic analyses showed higher expression of tetracycline resistance genes..... which also agreed with our in vitro experiments with pure cultures of ruminal bacteria".*

Again, what do these suggest? Please indicate the nature of the metatranscriptome dataset and indicate future survey of expression of detected ARGs detected in this study under different feeding, environment, through different growth stage and so on. Also, the authors should also point out to survey these ARGs through GIT of cattle to indicate their potential transit through the gut by ICEs.

Response: The discussion has been modified to accommodate the reviewer's concerns and analyses for further studies were indicated, as suggested by the reviewer. As mentioned above, meta-data regarding most ruminal transcriptomes are lacking, making it difficult to provide details about the nature of the metatranscriptomes analyzed in this manuscript. We thank you again for such valuable suggestions and hope that the changes provided in the text meet the reviewer's concerns.

Reviewer #2 (Remarks to the Author):

The manuscript has been modified in a satisfactory way, and there is no more queries from my part.

Response: We thank the reviewer for the positive feedback about the manuscript and several useful suggestions that helped improve the paper.

Consider these minor changes:

Table 1: Please correct "Fusic acid". Should be "Fusidic acid"; "Oxazolidone" should be "Oxazolidinone", "Fenicol" should be "Phenicol". Better in plural: aminoglycosides, beta-lactams, sulphonamides, tetracyclines. Spell MLS: Macrolides, Lincosamides, Streptogramin B. Please apply these changes to the antibiotic designations at the end of page 21.

Response: We thank the reviewer for pointing this out. The text has been corrected as indicated. Thank you.

Page 6, fifth line from the bottom. Should be: "Bacteroides ovatus.... was the only species in Bacteroidetes possessing genes for sulphonamide resistance". Check if that is right as I'm writing it.

Response: The reviewer is correct. The text has been corrected as suggested. Thank you.

Two lines below: "Proteus mirabilis"

Response: The reviewer is correct. The text has been corrected as suggested. Thank you.